



# Quantifying frost weathering induced rock damage in high alpine rockwalls

Till Mayer[1,2], Maxim Deprez[3], Laurenz Schröer[3], Veerle Cnudde[3,4], Daniel Draebing[5,1]

[1] Chair of Geomorphology, University of Bayreuth, 95447 Bayreuth, Germany.

[2] Chair of Landslide Research, Technical University of Munich, 80333 Munich, Germany

[3] Department of Geology, University Gent, 9000 Gent, Belgium.

[4] Department of Earth Sciences, Utrecht University, 3584 CB Utrecht, Netherlands.

[5] Department of Physical Geography, Utrecht University, 3584 CB Utrecht, Netherlands.

*Correspondence to*: Till Mayer (till.mayer@uni-bayreuth.de)

**Abstract.** Frost weathering is a key mechanism of rock failure in periglacial environments and landscape evolution. At high alpine rockwalls, freezing regimes are a combination of diurnal and sustained seasonal freeze-thaw regimes and both influence frost cracking processes. Recent studies have tested the effectiveness of freeze-thaw cycles by measuring weathering proxies for frost damage in low-strength and grain-supported pore space rocks, but detecting frost damage in low-porosity and crack-

dominated alpine rocks is challenging due to small changes in these proxies that are close to the detection limit. Consequently, the assessment of frost weathering efficacy in alpine rocks may be flawed. In order to fully determine the effectiveness of both freezing regimes, freeze-thaw cycles and sustained freezing were simulated on low-porosity high-strength Dachstein limestone under temperature and moisture conditions that reflect those found in high alpine rockwalls. Frost-induced rock damage was uniquely quantified by combining X-ray computed micro-tomography (μCT), acoustic emission (AE) monitoring and frost

cracking modelling. To differentiate between potential mechanisms of rock damage, thermal- and ice-induced stresses were simulated and compared with AE activity. μCT combined with AE data revealed frost damage on low-porosity alpine rocks with crack growth along pre-existing cracks with magnitudes dependent on the initial crack density. It was observed that diurnal freeze-thaw cycles have a higher frost cracking efficacy on alpine rocks compared to a seasonal sustained freezing regime. On north-facing high alpine rockfaces, the number of freeze-thaw cycles and the duration of sustained freezing

conditions vary with elevation and seasonal climate. The experimental results establish a link between frost damage and elevation-dependent rockwall erosion rates, which has implications for hazard prediction in mountainous areas under a changing climate.

**Keywords.** Frost weathering, micro-CT, periglacial processes, frost cracking modelling, rock damage




## 1 Introduction

Frost weathering is a key mechanism for rock breakdown in periglacial environments (Fig. 1; Matsuoka and Murton, 2008) and is therefore considered to be the main driver for alpine landscape evolution (Egholm et al., 2015; Hales and Roering, 2009). The breakdown of rock due to freezing is called frost cracking and can prepare and trigger rockfall (Matsuoka, 2019).

Cracking can occur when stresses exceed thresholds (critical cracking) or by repetition of low magnitude stresses that progressively weaken the rock (subcritical cracking; Eppes and Keanini, 2017). Frost cracking was associated with two different processes: volumetric expansion and ice segregation (Matsuoka and Murton, 2008). Volumetric expansion of 9% occurs when water freezes to ice and can theoretically build up stresses of up to 207 MPa (Matsuoka and Murton, 2008), which would exceed the tensile stress of most existing rock (Perras and Diederichs, 2014). Conditions that favour volumetric

expansion are a high degree of water saturation of 91 % (Walder and Hallet, 1986), a rapid freezing associated with diurnal freeze-thaw cycles (Matsuoka, 2001; Matsuoka and Murton, 2008) and a freezing from all sides (Matsuoka and Murton, 2008). However, rock moisture conditions exceeding 91 % are rarely given in alpine rockwalls (Sass, 2005a) during time periods of diurnal freezing, e.g. in autumn or spring or during sustained freezing in winter (Fig. 2b). In addition, topography influences the temperature regime of rockwalls and therefore their freezing behaviour (Noetzli et al., 2007), making freezing from all

sides unlikely. Therefore, the conditions for volumetric expansion are unlikely in alpine rock faces, and any stresses inducing rock failure may result from ice segregation. Laboratory studies have observed a decrease in elastic and mechanical properties after frequent freeze-thaw cycles (Eslami et al., 2018; Jia et al., 2015), which has been interpreted as rock fatigue or subcritical cracking due to volumetric expansion. However, these studies do not provide insight into the timing of cracking and cannot exclude the influence of alternative cracking processes such as thermal stress (Eppes et al., 2016) or ice segregation by water

migration towards ice crystals (Prick, 1995; Prick et al., 1993).

Under freezing conditions, an ice crystal forms in cracks and pores. Due to repulsive forces a thin (<9 nm) film of water remains between the ice crystal and the pore or crack wall (Gilpin, 1979; Webber et al., 2007; Sibley et al., 2021), allowing water flow driven by a thermodynamic potential gradient from either unfrozen rock or unfrozen water within frozen rock

towards the ice crystal (Derjaguin and Churaev, 1986; Kjelstrup et al., 2021; Everett, 1961; Gerber et al., 2022). This process is called ice segregation and the pressure generated by the growth of ice in pores and cracks is called crystallisation pressure (Scherer, 1999) and is the main driver of ice segregation induced rock failure, known as frost cracking. Ice segregation can occur in low-saturated rock (Mayer et al., 2023) and is associated with seasonal freezing (Matsuoka and Murton, 2008). Murton et al. (2006) demonstrated that the presence of permafrost enhances ice segregation by refreezing rock moisture at the

permafrost table caused by active-layer thawing. The temperature range in which ice segregation is most efficient is called the 'frost cracking window' (Anderson, 1998) and depends on rock strength (Walder and Hallet, 1985; Mayer et al., 2023). Common temperature ranges vary from -6 to -3 °C for low-strength Berea sandstone (Hallet et al., 1991), but can drop to -15 to -5°C for high-strength rocks (Walder and Hallet, 1985; Mayer et al., 2023). Therefore, ice segregation is theoretically





favoured by low freezing rates and sustained freezing temperatures occurring during seasonal freezing (Walder and Hallet, 1986).

In alpine catchments, frost cracking activity is difficult to assess. Girard et al. (2013) used acoustic emission (AE) to monitor frost cracking, but other thermal stress-induced cracking processes (Eppes et al., 2016) cannot be excluded as alternative sources of AE signals. Most commonly, frost cracking is modelled using simulations with a fixed frost cracking window between -8 and -3 °C (Hales and Roering, 2007; Anderson et al., 2013) or a rock strength dependent frost cracking window (Rempel et al., 2016) and applying an elevation dependent air temperature as a proxy for rockwall temperatures. However, topographic effects in mountain rockwalls affect insolation by exposition and snow cover duration (Haberkorn et al., 2015a; Draebing et al., 2017b; Hasler et al., 2011; Magnin et al., 2015), therefore, rock surface temperatures differ from air temperatures in the magnitude of temperatures and frequency of freeze-thaw cycles. To account for topographic effects, previous studies have used recorded rock temperatures to drive frost cracking models (Sanders et al., 2012; Draebing and Mayer, 2021) and validated models with fracture spacing measurements (Draebing and Mayer, 2021) or rockwall erosion rates derived by terrestrial laserscanning (Draebing et al., 2022). With increasing elevation and hence colder prevailing temperatures, frost cracking models indicate an increase in frost cracking activity up to an elevation range, where temperatures are sufficiently cold to limit water transport towards ice lenses (Draebing et al., 2022; Rempel et al., 2016). These elevations are characterised by long periods of sustained freezing (e.g. Matterhorn, Aiguille di Midi in Fig. 1a-b and Fig. 2a), in contrast to lower elevation rockwalls where frequent freeze-thaw cycles are common (e.g. Dammkar in Fig. 1e and Fig. 2a).



**Figure 1: Selected alpine rockwalls a) Aiguille du Midi (Mont Blanc Massif, French Alps), b) Matterhorn (Valais, Swiss Alps) c) Corvatsch (Engadin, Swiss Alps),  d) Gemsstock (Uri, Swiss Alps), e) Dachstein (Northern-Calcareous Alps, Austria) and f) Dammkar (Northern-Calcareous Alps, Germany) used for the analysis of freezing regimes in Fig. 2.**




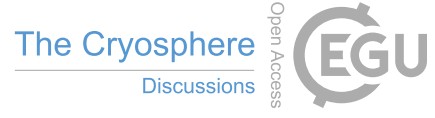

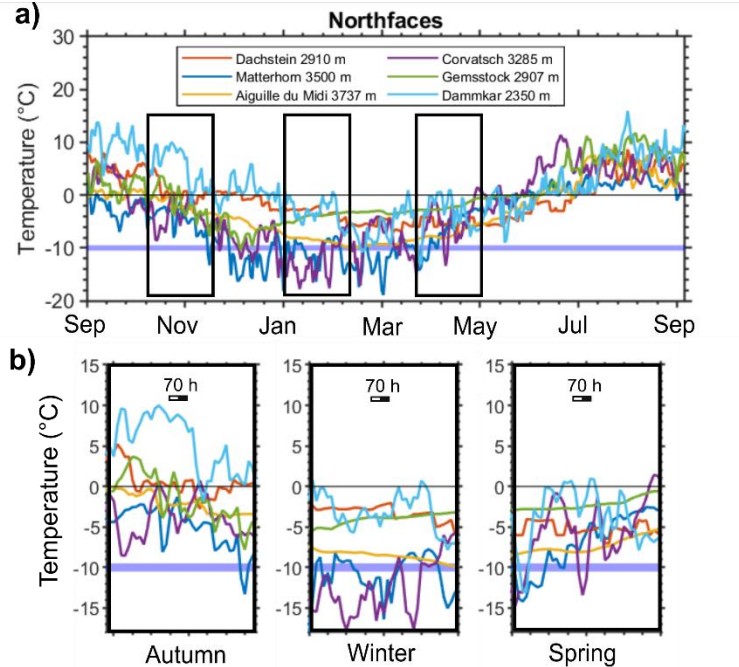

**Figure 2: a-b) Diurnal freeze-thaw cycles and sustained freezing based on measured rock surface temperatures in 10 cm depth at north-facing rockwalls for Matterhorn 2017/18 (Weber et al., 2019), Aiguille du Midi 2014/15 (Magnin et al., 2015), Corvatsch**
**2020/21 (Permos, 2022), Gemsstock 2017/18 (Permos, 2022), Dachstein and Dammkar 2020/21 (unpublished own data). The lilac bold line indicates -10° C, the applied lower rock temperature boundary of our experimental set up.**

Draebing and Krautblatter (2019) quantified ice-induced stresses by volumetric expansion and ice segregation in rock samples with a mode I cracks, and describe the mechanisms of ice-induced fracture opening in the field (Draebing et al., 2017a). The

stresses generated by ice segregation were below the rock strength threshold and therefore in the subcritical cracking range (Eppes and Keanini, 2017). Currently, stresses can be calculated using mechanical models, or rock damage can be estimated using proxies such as decreases of p-wave velocity or Young's modulus or small increases in porosity (Whalley et al., 2004; Matsuoka, 1990; Prick, 1997). However, in low-porosity rocks, changes in frost cracking proxies are very small and often within the uncertainty of the techniques used, and therefore do not provide reliable results. To our knowledge, no study has

directly quantified rock damage and hence the effectiveness of freeze-thaw cycles in inducing frost cracking. Furthermore, laboratory freeze-thaw tests have never demonstrated whether frost cracking creates new cracks or propagates existing cracks in high-strength, intact, low porosity rock.

In this study, we tested the efficacy of frequent diurnal and sustained seasonal freeze-thaw cycles reflecting temperature conditions in high alpine rockwalls (Fig. 2a-b) on low-porosity, high-strength Dachstein limestone. For this purpose, we

conducted laboratory freezing experiments while monitoring acoustic emission events and modelling thermal and ice-induced stresses. To assess frost cracking efficacy, we applied X-ray computed micro-tomography (µCT) to identify and quantify crack propagation.





## 2 Material and Methods

### 2.1 Rock samples and mechanical properties

We collected three large rock boulders from a scree slope adjacent to a north-facing rockwall located at 2650 m in the Dachstein mountain range, Austria. Dachstein limestone (upper Triassic) is a massive rock with minor occurrence of fractures. We drilled three cylindrical samples with a size of 10 x 5 cm and measured rock density $\rho_r$ and open porosity $n_r$ by immersion weighing (DIN 52102 and DIN EN 1097-6). The determined rock density was 2690 kg m$^{-3}$ and rock porosity 0.1% (Table 1). To quantify shear modulus $G$, Poisson' ratio $v$ and Young's modulus $E$, dilatational wave velocity (VD) measurements were performed using a Geotron ultrasonic generator USG40 and a Geotron preamplifier VV51 with 20 kHz sensors. Sensor to sample coupling was improved by applying 0.2 MPa pressure. Detection and analyses of the signals were proceeded with a PICO oscilloscope and the software Geotron Lighthouse DW. Determined values of shear modulus were 24.02 ±0.05 GPa, Poisson' ratio 0.327 ±0.003 and Young's modulus 63.74 ±0.04 GPa (Table 1). We drilled two cylindrical samples with a size of 5 x 2.5 cm to measure tensile strength $\sigma_t$ following Lepique (2008). The determined tensile strength was 7.9±0.7 MPa. Fracture toughness $K_{iC}$ was tested in the Magnel-Vandepitte Laboratory at Ghent University using a three-point bending (TPB) test setup (Carloni et al., 2019; Bazant and Planas, 1998). For Dachstein limestone the fracture toughness was 1.32±0.1 MPa m$^{1/2}$. All freezing experiments were conducted on six 91.5 x 25.5 mm large cylindrical samples (Fig. 3a) drilled from the same rock boulder.

**a)**

Freeze-Thaw cycles (FT-1)   Sustained Freezing cycle (FT-2)

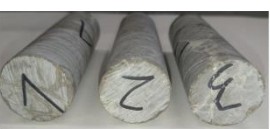

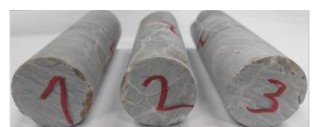

**b)**

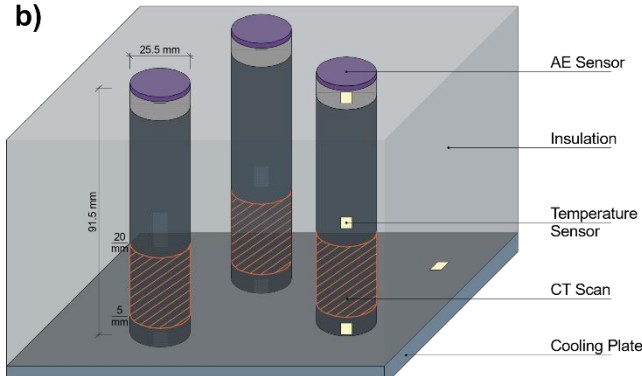





**Figure 3: a) 91.5 x 25.5 mm large cylindrical Dachstein limestone samples used for freeze-thaw experiments. b) Schematic representation of freezing laboratory setup. We established a temperature gradient by cooling three rock samples with different degrees of saturation with a cooling plate at the bottom while the top of these samples was exposed to room temperature conditions.**

**2.2 Freezing experiments setup**

We tested six samples (Fig. 3a) in two different experiments in which three samples were exposed to multiple freeze-thaw cycles (FT-1) and three samples to a sustained freezing cycle (FT-2, see sites with temperatures below 0°C in Fig 2a). Saturation of the samples was conducted by immersing the lower part of the rock samples into a distilled water bath. To prevent air inclusions, we raised slowly the water table until samples were completely immersed, and a constant mass was prevailing (we refer to as 100 % or fully saturated). Subsequently, samples were dried under atmospheric conditions and weighed until a saturation of 30 % or 70 % was reached. To prevent rock moisture loss due to evaporation, the samples were wrapped in clingfilm during the experiment. The samples were placed into an insulating holder on a cooling plate, which was driven by Peltier elements (TE technology, CP-121HT), a power supply unit (PS-24-13), and a temperature control system (TC-48-20 OEM, Fig. 3b). While the bottom part was exposed to freezing conditions from the cooling plate, the top part was left open to non-freezing ambient laboratory conditions (20 °C). With this setup we simulated an open system that provides a linear temperature gradient and a water body inside the rock samples reflecting natural rockwall conditions. Frost cracking efficacy was tested by applying two different experimental cooling cycles. For the first freeze-thaw experiment (FT-1), the cooling plate was oscillating between -20 °C (2 hours) and 5 °C (1 hour). In total, the FT-1 lasted for 68 h and cycled 20 times between freezing and thawing conditions. In the second experiment, a sustained freezing cycle (FT-2) was applied by setting the cooling plate to -20 °C for 68 h. The cooling plate temperature of -20 °C provided a rock surface temperature at the bottom of the sample of -10 °C, which reflected north facing rockwall conditions (Fig. 2a).

During the freezing experiments, rock temperature was monitored every minute at the side close to the top, middle and bottom of one rock sample with three SE060 high temperature type K thermocouples (accuracy ±0.5 °C, Pico Technology, Fig. 3b). An additional thermocouple was placed onto the cooling plate to record cooling plate temperatures. To monitor acoustic emission (AE) which are used as a proxy for cracking (Eppes et al., 2016; Hallet et al., 1991), a Physical Acoustics AE sensor PK6I with a frequency between 35 and 65 kHz was mounted with acrylic sealant on top of each rock sample. The detected AE signals were recorded with a Physical Acoustics micro SHM node. Recorded data were subsequently processed and filtered using Physical Acoustics AEwin software. We used an initial signal threshold of 30 dB$_{AE}$ and lead break tests to evaluate wavelength form and test the setup. To prevent false AE signals by the setup itself we tested the system by running without freezing or changing the temperature.





### 2.3 μCT imaging

In order to identify crack locations and quantify crack growth, X-ray computed micro-tomography (μCT) was performed at the Ghent University Centre for Tomography (UGCT) with the CoreTom (TESCAN XRE) μCT scanner. Only the bottom part 160 (0-20 mm) of the sample was scanned, since only this part experienced sub-zero temperatures (Fig. 3b). For experiment FT-1, μCT scans were performed before the start and after every five cycles (Fig. 4). For the sustained freezing experiment (FT-2), rock samples were scanned before and after the experiment. The μCT system settings for our experiment were set to 179 kV for the X-ray source with a power output of 20 W. The scans were performed at binning 2 and a voxel size of 20 μm. A 1 mm thick aluminium plate was used to filter low energy X-rays and reduce beam hardening. For each sample 2142 projections 165 were made with an exposure time of 600 ms. The raw μCT data was reconstructed using the software PANTHERA (TESCAN XRE) where beam hardening and ring filters were applied, and which resulted in a stack of cross-sections saved as 16-bit tiff. images. All subsequent image handling, such as registration, segmentation, and analyses, were performed with Avizo3D Pro (Version 2021.1, ThermoFisher Scientific). In Avizo, a sandbox filter was conducted to bin contrast variations inside the images and match contrast between the single scans. We tuned the parameters until visually the best result was observed. 170 Therefore, sample voids (pore space) and matrix (sample material) of each image could be separated by thresholding over contrast. We followed the work after Deprez et al. (2020a) and defined a distinguishable feature in the scan image as a minimum spatial resolution of 3 times the voxel size (60 μm). Volume fractions were determined for each image by calculating differences between voids and matrix which we used for later comparison between the scans to evaluate pore space growth. The parameter crack volume fraction was defined by the total amount of segmented pore space (voids) per image in the image 175 stack (cross section) divided by the total amount of segmented sample material (matrix). Due to beam hardening effects between 0 and 5 mm as well as 19 and 20 mm rock sample height (from the bottom), we focussed in our analyses on 5 to 19 mm height of our rock sample. We quantified crack growth (pore space growth) by comparing the crack volume fraction per layer after each scan (Fig. 6a-f). Potential influences of initial varying crack density were evaluated by normalizing each sample scan with its own initial crack volume (Fig. 6g-i).


### 2.4 Thermal- and ice-stress modelling

We modelled thermal and ice stresses to determine potential drivers of AE signals and porosity growth. The models require a one-dimensional temperature distribution inside the samples. Temperature distributions were calculated by assuming a linear and homogeneous temperature gradient between all temperature sensors (Fig. 3b). Latent heat effects were incorporated in 185 general as our sensors measured rock temperatures which were affected by latent heat effects. All thermal and ice stress simulations were performed in MATLAB (2021). Thermal stress occurred in our samples as a result of changing temperatures. We modelled one-dimensional thermal stress $\sigma_{TL}$ after Paul (1991):

$$\sigma_{TL}(t) = 2 \cdot G \cdot \alpha_T \left| \frac{dT_{L\pm s}}{dt} \right| (1 + \nu), \tag{1}$$



with thermal expansion coefficient $\alpha_T$, shear modulus $G$ and Poisson ratio $v$ (Table 1). $T_{L\pm5}$ represents a running temperature

mean over five minutes. Our model approach did not incorporate complex crack geometries of the samples; therefore, we could only provide quantitative estimates of thermal stress. Hence, we focused in our analysis on timing of thermal stresses and did not analyse absolute stress values.

We applied frost cracking modelling to determine the time dependency of ice stresses during the test. Frost cracking modelling

was performed using the model of Walder and Hallet (1985), which combines hydraulic and mechanic rock properties. The model simulated ice pressures in a single 1 mm long mode I crack. Therefore, this model simplified crack geometry and provided quantitative estimates of ice pressures. Due to the abstract model predictions, we used quantified ice pressures only to interpret timing of AE events and not as absolute values.

The basic model requirement to start ice segregation are rock temperatures below the pore freezing point $T_f$ and an unfrozen area inside the rock, which acts as a water reservoir. The transition area in between a potential ice lens in the frozen part and the unfrozen area is called frozen fringe. The model assumes a fully saturated rock. Ice pressure rises between lens and pore/crack wall when water migration is driven by a thermodynamic potential gradient through the frozen fringe. The amount of supplied water is governed by Darcy's law but restrained due to hydraulic conductivity inside the frozen fringe and flow

resistance due to the thin film between ice and crack wall.

Following Walder and Hallet (1985), we chose a hydraulic conductivity of $k_{hc}$ of $5\times10^{-14}$ m s$^{-1}$. Flow resistance between the ice and pore wall depends on grain size $R$, liquid-layer thickness $h_l$, and ice-free porosity $n_f$. Walder and Hallet (1985) set the grain size to 0.75 mm, the liquid-layer thickness to 6 nm °C$^{1/2}$ (after Gilpin (1980) at a temperature of -1 °C) and neglected

ice-free porosity. A simplified potential ice lens has the form of a penny shaped crack with a crack radius $c$ and a width $w$. With water migrating towards the ice lens, ice pressure rises inside the lens and finally leads to tip cracking (mode I type). The initial crack length (two times the crack radius) is set to 1 mm with crack orientation parallel to the sample bottom ($\varphi_p$=0 °). After Walder and Hallet (1985) shear modulus $G$ and Poisson' ratio $v$ determine how the penny shaped crack is deformed elastically into an oblate ellipsoid when ice pressure is applied and be described for very thin cracks (w<<c) to

$$\frac{w(n,t)}{c(n,t)} = \frac{4}{\pi}\left(\frac{1-v}{G}\right)p_i \tag{2}$$

The crack finally breaks subcritical inelastically at the tip (mode I type) and propagates crack growth $V$ when one third of the critical fracture toughness $K_C$ ($K_* = 1/3K_c$) is exceeded, for $K_I>K_*$ crack growth can be expressed after Walder and Hallet (1985) as:

$$V = V_c[e^{\gamma(\frac{K_I^2}{K_c^2}-1)} - e^{\gamma(\frac{K_*^2}{K_c^2}-1)}], \tag{3}$$






with $K_I$ being the stress intensity factor and V=0 when $K_I \leq K_*$. Subcritical cracking can be also expressed by the critical ice pressure (13.7 MPa) which is derived from transforming $K_I=(4c/\pi)^{1/2}p_i$ from Walder and Hallet (1985) and incorporating the measured critical fracture toughness of 1.32±0.1 MPa m$^{1/2}$ and an initial crack length of 1 mm. We set the dependent growth law parameters $V_c$ and $\gamma$ to be 340 m s$^{-1}$ and 37.1 m s$^{-1}$ (Table 1) after Westerly granite (Walder and Hallet, 1985). We are

aware that this model only reflects partly the properties of our rock samples as crack length (or orientation) and hydraulic conductivity vary between the samples. Previous sensitivity analyses revealed that chosen crack length shifted frost cracking magnitude and also slightly the crack timing but it did not affect the overall frost cracking pattern in terms of location (Draebing and Mayer, 2021). We followed studies by Walder and Hallet (1985) and Draebing and Mayer (2021) which produced realistic results for our chosen parameters.


**Table 1. Parameters used for thermal stress and frost cracking modelling of Dachstein Limestone samples.**

| Parameter | | Value |
|---|---|---|
| Ice density (kg m$^{-3}$) | $\rho_i$ | 920 |
| Water density (kg m$^{-3}$) | $\rho_w$ | 1000 |
| Pore freezing point (°C)[*] | $T_f$ | -1 |
| Hydraulic conductivity (m s$^{-1}$)[*] | $k_{hc}$ | $5\times10^{-14}$ |
| Grain size (mm)[*] | R | 0.75 |
| Liquid layer thickness (nm °C$^{1/2}$)[*] | $h_l$ | 6 |
| Initial crack radius (m)[*] | $x_i$ | 0.005 |
| Angle between crack plane and rock wall (°)[*] | $\phi$ | 0 |
| Poisson' ratio () | $\nu$ | 0.327 |
| Critical fracture toughness (MPa m$^{1/2}$) | $K_C$ | 1.32 |
| Growth-law parameter (m s$^{-1}$)[*] | $V_c$ | 340 |
| Growth-law parameter ()[*] | $\gamma$ | 37.1 |
| Rock density (kg m$^{-3}$) | $\rho_r$ | 2690 |
| Rock porosity (%) | $n_r$ | 0.1 |
| Shear modulus (GPa) | G | 24.02 |
| Young's modulus (GPa) | E | 63.74 |
| Thermal expansion coefficient (°C$^{-1}$)[+] | $\alpha_T$ | $6\times10^{-6}$ |

[*] Walder and Hallet (1985), [+] Pei et al. (2016)



# 3 Results

## 3.1 Results of AE logger and simulations

### 3.1.1 Freeze-Thaw cycles (FT-1)

The cooling plate exposed the bottom of the rock samples to temperatures oscillating between -19 and 6 °C. The temperature loggers between scan 0 and 1 were poorly coupled to the rock, which led to an offset between rock surface temperatures at the bottom between freeze-thaw cycles 1-5 compared to temperatures between freeze-thaw cycles 5-20 (Fig. 4a). The temperature sensor on the cooling plate was added after five freeze-thaw cycles (Fig. 4a). Between freeze-thaw cycles 5 and 20 the bottom rock temperature reached minimal temperatures of -10 to -8 °C and maxima of 8.5 to 9.5 °C. The upper temperature sensor showed overall positive temperatures varying between 15 and 24 °C. We detected 317 AE hits in the 100 % sample, 415 in the 70 % sample and 180 in the 30 % sample (Fig. 4c). During freezing phases more AE hits were detected compared to non-freezing phases. Most of the hits were detected when the bottom sensor showed temperatures below the freezing point with 235 hits for the fully saturated sample, 326 hits for the partly saturated and 123 hits for the low saturated sample. When the bottom sensor measured positive temperatures, we recorded 82 hits at the 100 % rock, 89 hits at the 70 % rock and 57 hits for the 30 % rock. The AE logger system stopped recording twice, between 17-19.5 h and 41.5-51 h, therefore, we underestimate the number of accumulated hits for the FT-1 cycle. Most AE hits were measured between scan 0 and 1 with a maximum of 121 hits for the 100 % sample, followed by 73 hits for the 70 % and 51 hits for the 30 % saturated rock sample. In the last period between scan 3 and 4, the 70 % saturated sample showed the highest AE accumulation with 173 hits, while sample 100 % had 53 hits and the 30 % only 44 hits.

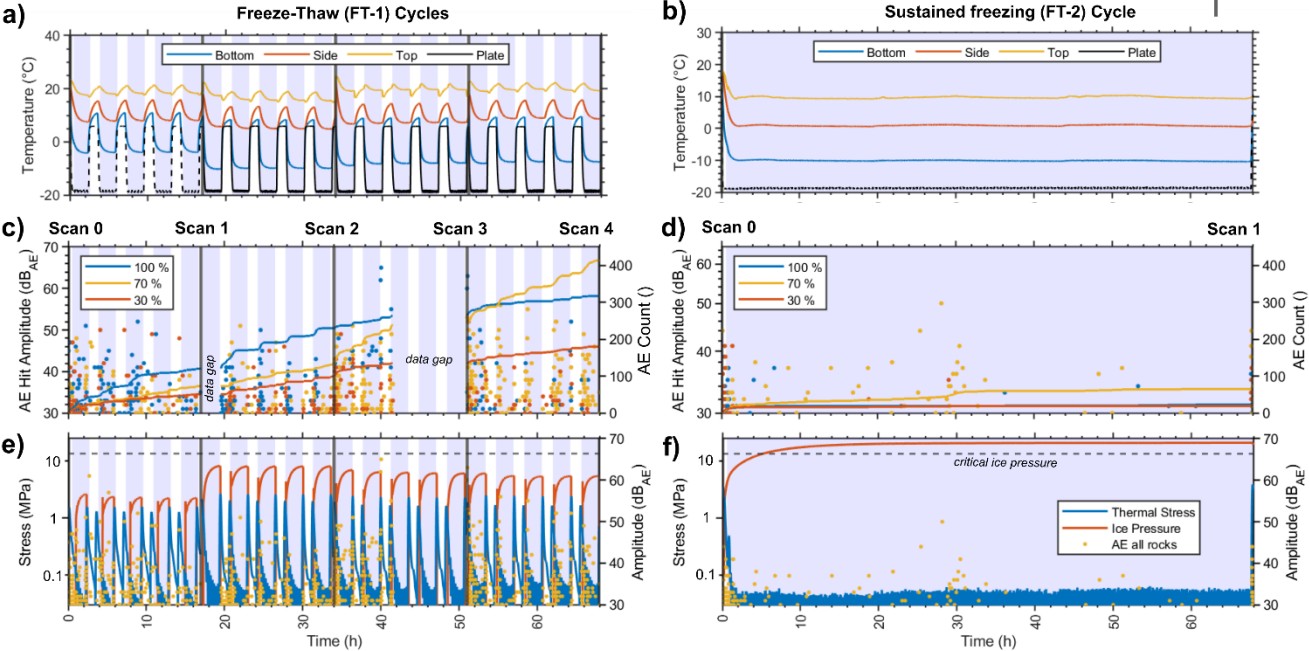

**Figure 4: a-b) Respectively measured rock and cooling plate temperatures, c-d) recorded AE hits and e-f) modelled thermal and ice stresses plotted against time for FT-1 and FT-2 with AE hits from all samples. The dashed black line highlights cooling plate temperatures according to the controller while black lines indicated measured plate temperature. The temperature offset between scan 0 and 1 during FT-1 (a) is a result of poor connectivity of the bottom temperature sensor. Blue rectangles highlight periods when bottom rock samples were exposed to temperatures below 0 °C.**

Our modelling yielded a maximum occurrence of thermal- and ice-induced stresses during freezing temperatures (Fig. 4e). The temperature offset between freeze-thaw cycles 1-5 resulted in an underestimation of modelled thermal and ice stresses (Fig. 4e). During freeze-thaw cycle 5 to cycle 20, peak thermal stresses increased when temperature shifted from thawing to freezing conditions with resulting pressures of 2.15±0.25 MPa and 2.95±0.20 MPa from freezing to thawing. At stable temperatures, thermal stress reached its minimum. Ice stress only occurred during freezing phases. Inside the freezing regime the model predicted rising ice stresses with maximum of 6.85±1.35 MPa at the end of each freezing phase, which is far below threshold for critical ice pressure (13.7 MPa) that can be referred to subcritical cracking.

**3.1.2 Sustained freezing cycle (FT-2)**

Rock samples were exposed to freezing conditions for 68 h with 66 h being at a fixed bottom rock temperature of -10±0.5 °C (Fig. 4b). During fixed temperature conditions the middle temperature sensor stayed at positive temperatures of 0.9±0.3 °C and the top sensor at 9.8±0.4 °C. Our AE loggers detected 28 hits at the 100 % saturated, 77 hits at the 70 % saturated and 23 hits at the 30 % saturated rock sample (Fig. 4d). The fixed freezing phase caused 11 hits to occur at the fully saturated rock,





53 hits at the partly saturated rock and 8 hits at the low saturated rock. Thermal stress modelling predicted peak values of 3.6
MPa for cooling and 3.9 MPa for warming at the beginning and end of the freezing cycle (Fig. 4f). During fixed temperatures
thermal stress stayed below 0.1 MPa. Throughout the freezing phase the ice stress model predicted a rising ice stress with a
maximum of 21 MPa. Ice pressure exceeded the critical ice pressure of 13.7 MPa after 5.4 h.

## 3.2 Results of µCT

### 3.2.1 Freeze-Thaw cycles (FT-1)

As the experiments were executed with a spatial resolution of 60 µm the µCT scans show that most of the pore volume in our
samples are cracks and that volume changes occur in form of crack growth (Fig. 5). The data revealed that crack growth was
independent of the height in the sample, however, crack growth revealed a positive correlation between initial crack volume
and crack growth (Fig. 6a-c). The initial fraction of crack volume (volume of crack divided by total volume) per layer from
scan 0 before freezing varied between 0.014 – 0.042 for the 30 % rock, 0.003 – 0.011 for the 70 % rock and 0.005 – 0.012 for
the 100 % rock. The crack volume increased uniformly independently from height at all tests.

To exclude the sensitivity of crack growths to initial crack distribution, we normalized the crack volume growth for each
sample by its initial crack volume. Normalization revealed that crack volume growth was not dependent over the height of the
samples, however, the amount of crack volume growth per scan varies with saturation (Fig. 6g-i). For the whole experiment,
the final scan 4 showed a mean normalized crack growth of 1.34 ±0.8 (i.e. 34 % more crack volume than initially) for the 30 %
saturated sample, 1.29 ±0.11 (i.e. 29 % more crack volume than initially) for the 70 % saturated sample and 1.52 ±0.13 (i.e.
52 % more crack volume than initially) for the fully saturated sample. The average normalized growth of volume in between
scans was 0.8 ±0.03 (i.e. 8 % more crack volume than initially) for the 30 % rock, 0.07 ±0.04 (i.e. 7 % more crack volume
than initially) for the 70 % rock and 0.13 ±0.05 (i.e. 13 % more crack volume than initially) for the 100 % rock.




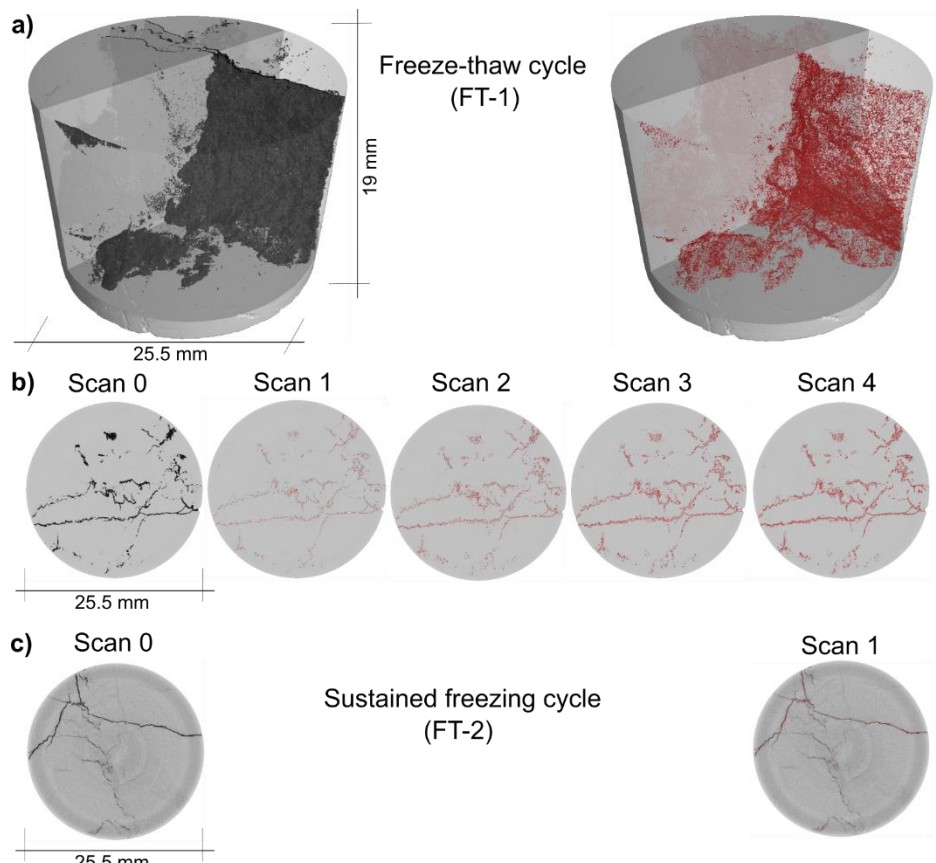

**Figure 5: a) 3D CT scans before (scan 0) and after the last freeze-thaw cycle (scan 4) of low-saturated (30 %) rock samples experiencing FT-1. CT scan slices at 8 mm depth from the bottom for b) the 30 % saturated sample exposed to FT-1 and c) for the 70 % saturated samples experiencing FT-2.**







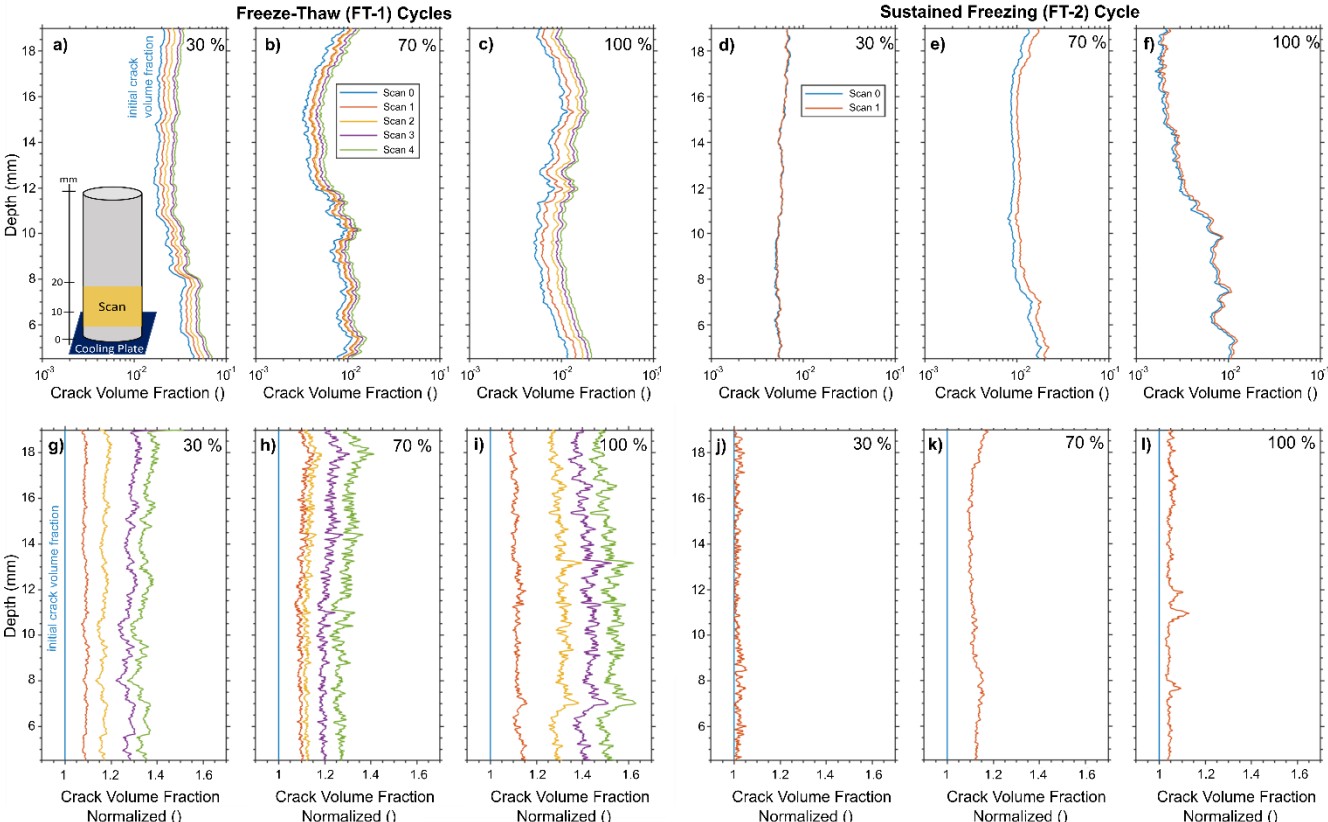

**Figure 6: a-f) Measured crack volume fraction (volume of cracks divided by total volume) using µCT plotted against rock depth. g-l) Quantified normalized crack volume fraction by initial crack volume (blue line a-f) plotted against rock depth.**

### 3.2.2 Sustained freezing cycle (FT-2)

The sustained freezing experiment revealed that the highest crack volume growth occurred in the 70 % saturated rock sample, followed by the fully saturated rock sample, while the 30 % saturated rock sample showed almost no crack growth (Fig. 6d-f). Scan 0 showed an initial crack volume fraction between $0.005 - 0.008$ for the 30 % saturated rock, $0.008 - 0.017$ for the 70 % and $0.001 - 0.011$ for the fully saturated rock. Normalization revealed the highest crack growth at the partly saturated sample (Fig. 6j-l). We calculated a mean normalized crack growth for the 30 % rock with $1.02 \pm 0.02$ (i.e. 2 % more crack volume than initially), for the 70 % rock with $1.12 \pm 0.04$ (i.e. 12 % more crack volume than initially), and for the 100 % rock with $1.05 \pm 0.03$ (i.e. 5 % more crack volume than initially).




## 4 Discussion

### 4.1 CT technique and crack history

In alpine rocks, freezing leads to pore-space growth along pre-existing cracks, which is influenced by the initial crack density. Our normalised µCT results revealed a uniform increase in pore space growth along the depth of the sample. This pattern suggests an influence of crack density on crack growth with increased crack growth due to higher initial crack density or initial pore volume (Fig. 6g-i). This is supported by the scan visualization (Fig. 5), which highlights the pore space growth mainly in pre-existing cracks. Our results are consistent with theoretical assumptions by Scherer (1999) or Walder and Hallet (1985) that ice crystallisation pressure increases in pore spaces, which is supported by microscopic findings by Gerber et al. (2022) that can lead to pore or crack growth. Previous µCT measurements on highly porous (30-40 %) oolithic limestone building stone by Deprez et al. (2020a) or highly porous (~35 %) miliolid limestone by De Kock et al. (2015) showed the microscopic processes occurring in limestone pores during freeze-thaw cycles. The authors were able to relate the pore processes to possible macroscopic damage patterns. Both experiments used small samples with 8-9 mm in diameter, allowing a resolution of 21 µm (Deprez et al., 2020a) and 20.4 µm (De Kock et al., 2015). As we aimed to generate a thermal gradient and water migration, we used larger samples resulting in a lower spatial resolution of 60 µm.

Our experiments have shown that we can visualize and quantify freeze-thaw induced rock damage in low-porosity rock samples, considering potential small additional damage that may be below µCT resolution. Previous laboratory freeze-thaw tests have used a decrease in elastic properties such as P-wave velocity, Young's modulus or increases in porosity (Whalley et al., 2004; Matsuoka, 1990; Draebing and Krautblatter, 2012) as a proxy for frost damage. However, in low-porosity alpine rock frost damage is difficult to detect because changes in elastic properties or porosity are small or below the level of detection. Furthermore, the use of measured elastic properties or porosities does not provide information on crack geometry and crack growth. We demonstrated that cracks grow along pre-existing cracks and showed that that µCT is a powerful analytical tool.

### 4.2 Potential thermal and ice stress sources inducing cracking

Thermal and ice stresses or a combination of these stresses can cause rock damage. We monitored AE as a proxy for cracking as previous stress experiments (Eppes et al., 2016; Hallet et al., 1991) and analysed the timing of AE events in combination with simplified thermal stress and ice stress models to decipher the potential stress source.

Cooling of the samples induced thermal stresses (Fig. 4e-f) and triggered AE events (Fig. 4c-d) which we interpret as crack propagation similar to previous studies (Eppes et al., 2016; Collins et al., 2018; Mayer et al., 2023). When temperatures dropped below 0 °C, freezing enabled the development of ice stresses which could have amplified thermal-induced crack propagation. According to the volumetric expansion theory, ice stresses should develop in samples with a saturation higher than 91% (Walder and Hallet, 1986). Moreover, Prick (1997) empirical findings suggested the existence of a saturation threshold between 58 and 100 % for the occurrence of volumetric expansion or ice segregation. However, measurements were





conducted mostly on highly porous homogeneous limestone samples with porosities between 9.15 and 47.23 % which differ strongly to our low porous and crack dominated alpine rock. Only the Comblanchien limestone with a porosity of 1.3% was

in the range of our  samples porosity and showed a critical saturation threshold of 100 % (Prick, 1997). In addition, all rocks tested by Prick (1997) had an intergranular porosity which differs to our fracture-controlled alpine rock samples which impacts fracture propagation behaviour (Deprez et al., 2020b). We conclude that only our 100 % saturated samples fulfil the moisture conditions for volumetric expansion and consequently we cannot exclude the occurrence of volumetric expansion within these samples. However, while temperatures dropped below the freezing point in our experiments, our AE data showed no significant

differences in timing of AE signals between fully saturated samples and partially (70 %) or low-saturated samples (30 %). Ice segregation simulations during cooling revealed a slight increase of ice stresses which were nevertheless still below the critical threshold (13.4 MPa) for crack propagation for our samples (Fig. 4e,f). In conclusion, we interpret the AE activity as a result of thermal stresses during cooling and before the freezing point which was eventually amplified by ice segregation induced stressed at temperatures below the freezing point (Fig. 4c-f).


During further cooling of the rock samples, ice segregation causes crack propagation and therefore pore space growth. The highest increase of AE hits was observed during freezing at temperatures significantly below the freezing point (below -6 °C in FT-1 and FT-2, see Fig. 7) which we interpret as acoustic emission events caused by frost cracking induced crack growth. This temperature range is lower than the frost cracking window for Berea sandstone which lies between -3 and -6 °C observed

by (Hallet et al., 1991) but consists with findings of Mayer et al. (2023) on Wetterstein limestone (similar to Dachstein limestone) who suggested a maximum efficacy below -7 °C. The discrepancy can be explained by different pore geometrics (Deprez et al., 2020a; Mayer et al., 2023) of grain-supported Berea sandstone and fracture dominated Dachstein limestone and rock strength (Walder and Hallet, 1985; Mayer et al., 2023). The poor coupling of the bottom rock temperature sensor during the first five freeze-thaw cycles of FT-1 resulted in underestimation of rock temperatures (between -2 and -4 °C) that reduced

precited ice pressure by our frost cracking model. In general, the AE pattern is consistent with simulated ice pressure due to ice segregation that increase with decreasing freezing temperatures (Fig. 4e-f). Therefore, we suggest that unfrozen water migrated towards the freezing front as previously observed in laboratory studies (Prick, 1995; Prick et al., 1993) and caused crystallization pressures inducing cracking (Mayer et al., 2023). The highest increase of AE hits we observed in the partially saturated (70 %) rock samples in FT-1 and FT-2 resulting in the highest number of AE events. In contrast, the low-saturated

(30 %) samples showed the lowest number of AE hits in FT-1 and FT-2 which differs to findings of Mayer et al. (2023) who suggested no lower moisture boundary for ice segregation under the perquisite of available water within short distance (0.4 m).  However, our setup did not contain an external water bath like Mayer et al. (2023) which may not provide sufficient water for ice segregation in our low-saturated (30 %) samples. Based on AE activity our experiments suggest that higher saturation increases ice segregation. As our ice stress modelling assumes full saturation, we cannot test the effect of different saturation

regimes on model behaviour.

Compared to cooling, the warming stage caused less cracking indicated by AE events in FT-1 (Fig. 7a-c) as thermal stresses increased but ice stresses were reduced due to thawing. Previous laboratory measurements by Browning et al. (2016) revealed a maximum thermal stress during warming, however, the authors used much higher temperature ranges (up to 1000 °C). In

alpine rockwalls, Draebing (2021) observed more crack opening during cooling compared to warming and interpreted this pattern as a result of thermal stresses in combination with ice stresses. Our stress modelling (Fig. 4e) shows peak thermal stress with increasing ice stresses that support this argument and suggest a combination of thermal and ice segregation induced stress resulted in higher as expected AE hits during cooling cycles.

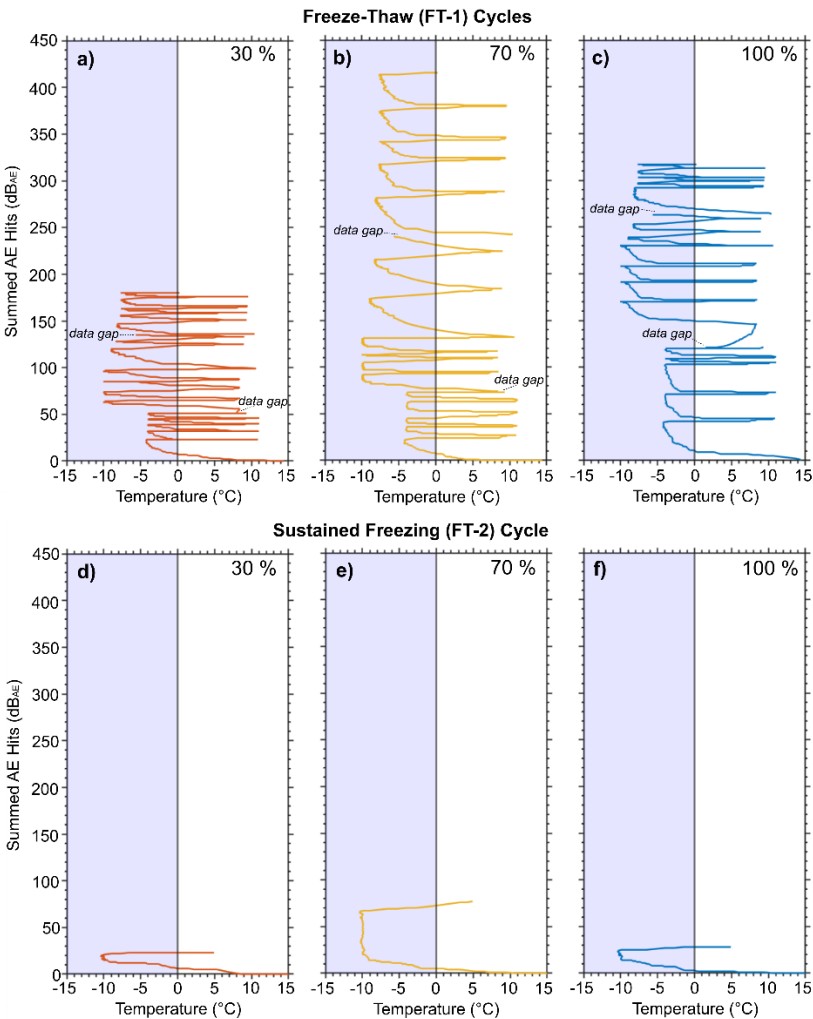

**Fig. 7: Cumulative AE hits in relation to bottom rock temperature for a-c) FT-1 and d-f) FT-2.**



### 4.3 Efficacy of freeze-thaw and sustained freezing cycles

Freeze-thaw cycles (FT-1) have a higher frost cracking efficacy compared to a sustained freezing-cycles (FT-1) in low-porosity crack-dominated alpine rocks. Absolute crack growth is affected by initial crack density or volume and cannot be compared directly; however, normalized crack growth fraction revealed an increase between 34 and 52 % (FT-1) compared to an increase

between 2 and 12 % (FT-2) (Fig. 8a, d). Based on μCT data, saturation levels have a minor impact on rock damage patterns, yet highest fracturing during FT-1 was observed in the fully saturated sample followed by the 30 % saturated sample in contrast to patterns observed by AE data (Fig. 8a). FT-2 revealed higher cracking in the 70 % saturated sample followed by the full saturation sample which is in accordance with AE data (Fig. 8d). We used AE data to determine the source of cracking by analysing the timing of AE hits. AE was used frequently as a proxy for thermal stress cracking in the field (Eppes et al., 2016;

Collins et al., 2018), frost cracking in the laboratory (Hallet et al., 1991; Maji and Murton, 2021) or field measurements (Amitrano et al., 2012; Girard et al., 2013). Our results show that the number of AE events and quantified rock damage by μCT are proportional (Fig. 8a,d), however, highest overall number of AE hits are not always in accordance with highest visible rock damage (Fig. 8a). The 70 % saturated sample revealed more than 415 AE hits with a normalized crack growth fraction of 47 %, while the 30 % saturated sample showed 180 AE hits and 53 % crack growth. This pattern could result from different

volume growth per crack propagation forcing less AE releases with higher crack growth. However, our natural rock samples may react in a different manner to stresses due to their slightly varying rock parameters which could affect number of AE hits. Predicted stresses by our models are higher for FT-2 in comparison to FT-1 which is not supported by our μCT findings. Our frost cracking modelling revealed eight times higher ice stresses in the sustained freezing phase than the freeze-thaw cycle (Fig. 8b,e). A perquisite for our frost cracking model is a full saturation which applies for the 100 % saturated samples in FT-

1 and -2. Fully saturated samples showed in contrast to our model predictions, that highest frost damage (+53 % more crack volume than initially) occurred during FT-1, whilst in FT-2 significantly less crack growth (+5 % more crack volume than initially) was observed. A reason for this pattern could be the occurrence of volumetric expansion that our frost cracking model does not incorporate. In addition, ice segregation requires water migration towards the freezing front that we provided by unfrozen water in the top of our samples. Prick (1997) observed water migration in limestone samples of comparable size with

porosities between 26 and 48.2 %. However, our sampled rocks have porosities of 0.1% which provide only a limited water reservoir potentially. Consequently, the water reservoir in FT-2 was the limiting factor for ice segregation despite temperature ranges favoring ice segregation (e.g. Walder and Hallet, 1985) and therefore, our ice stress simulations eventually overestimated. In addition, our frost cracking model simplified the complex crack geometries observed in the samples (Fig. 5) and simulated a single 1 mm long crack. Therefore, the magnitude of modelled ice stresses potentially deviates significantly

from occurred ice stresses which resulted in the deviation between modelled ice stresses and measured rock damage especially in FT-2 (Fig. 8b,e).



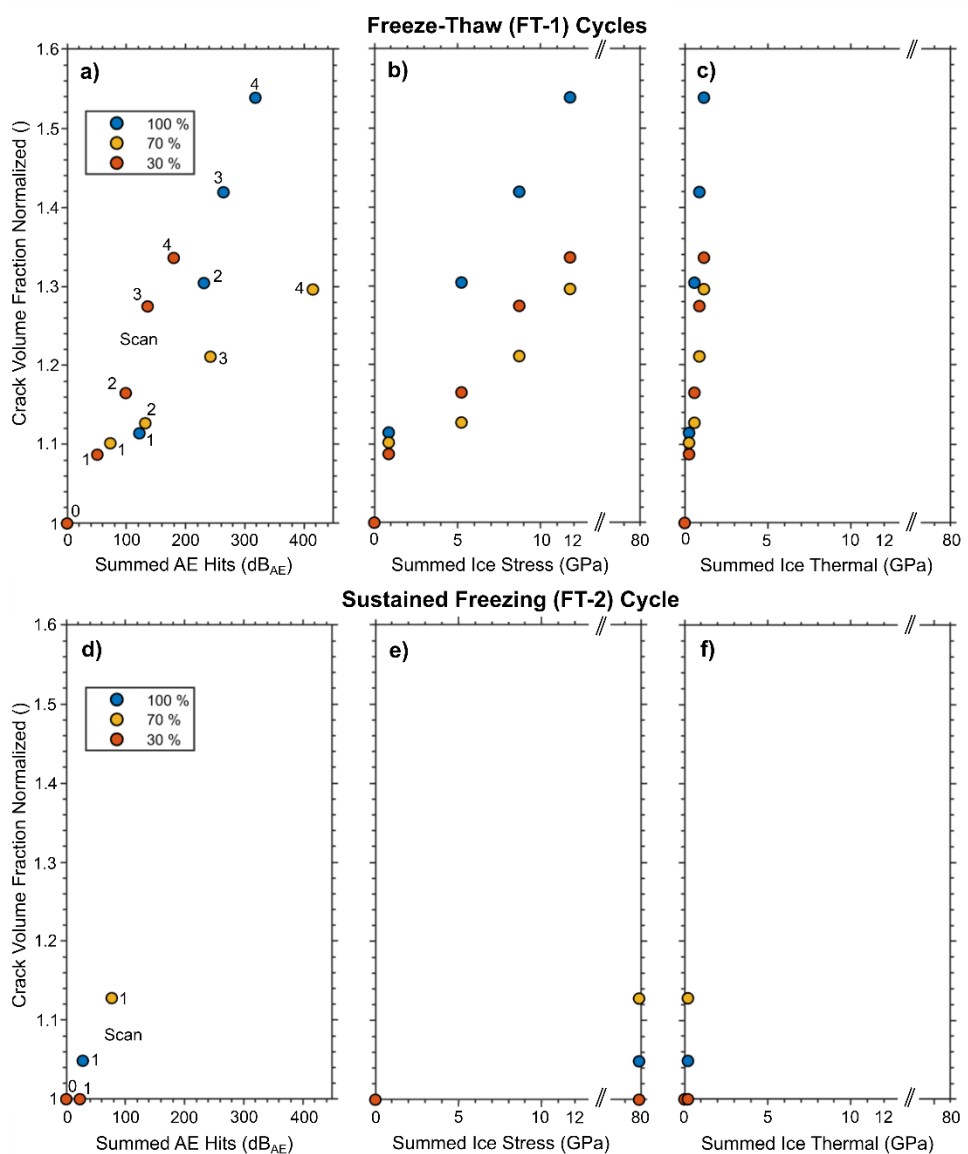

**Figure 8: Normalized crack volume fraction in relation to accumulated AE hits, simulated summed ice stress and thermal stress for a-c) FT-1 and d-f) FT-2.**


### 4.4 Implications for rock weathering in alpine rockwalls

The number of freeze-thaw cycles or sustained freezing periods within north-facing rockwalls is controlled by its elevation and snow cover (Fig. 2a). At elevations above 3000 m, as at Corvatsch, Matterhorn and Aiguille di Midi (Fig. 2a-b) sustained freezing periods are dominant throughout the year with likely permafrost occurrence (Gruber et al., 2004; Magnin et al., 2015;

Hasler et al., 2011). At elevations around 3000 m such as Dachstein or Gemsstock, the magnitude of freezing is decreased and



the length of the sustained freezing period is controlled by snow cover (Haberkorn et al., 2015a). These rockwalls are at the fringe of permafrost and show permafrost conditions in shaded rockwall areas (Haberkorn et al., 2015a; Haberkorn et al., 2015b; Rode et al., 2020). Snow duration decreases with elevation (Pepin et al., 2022; Pepin et al., 2015) and at lower elevation, such as Dammkar, sustained freezing is of short duration and the number of freeze-thaw cycles increase as temperatures

fluctuate between positive and negative rock temperatures especially in autumn and spring (Fig. 2b). A higher efficacy of freeze-thaw cycles than sustained freezing would result in higher erosion rates at low-elevation rockwalls, however, a requirement for reasonable frost damage are sufficiently cold temperatures to promote crack growth in high strength rock (Walder and Hallet, 1985) which is to some extent fulfilled at low elevated sites. In addition, Draebing et al. (2022) reviewed rockwall erosion rates in the European Alps and showed an elevation-dependent increase of erosion. The authors suggest that

the erosion rates increase as a result of frost cracking, permafrost and paraglacial processes.

South-facing rockwalls in the Alps experience more freeze-thaw cycles and less sustained freezing due to higher average rock temperatures and shorter snow duration. Previous studies observed an increase of mean annual rock surface temperatures in south-facing rockwalls compared to north-facing rockwalls between 2.2 to 6°C in the Hungerli Valley (Draebing and Mayer, 2021), between 3.3°C and 3.8°C at Gemsstock (Haberkorn et al., 2015a; Haberkorn et al., 2015b), up to 3.9°C at the Steintaelli

(Draebing et al., 2017b), up to 5°C at Aiguille du Midi (Magnin et al., 2015) and up to 7°C at Matterhorn and Jungfraujoch (Gruber et al., 2004; Hasler et al., 2011). Snow duration significantly decreases in south facing rockwalls (Draebing et al., 2017b; Haberkorn et al., 2015a; Haberkorn et al., 2015b) which results in an increase of freeze thaw cycles (Draebing and Mayer, 2021). A higher efficacy of frequent freeze-thaw cycles would result in more frost cracking at south-facing rock walls and higher erosion rates compared to north-facing rockwalls. Frost cracking modelling revealed more frost damage on south-

facing rockwalls in the Hungerli that authors attribute to high changes of thermal gradients (Draebing and Mayer, 2021). The authors suggested that the frost cracking magnitudes are overestimated due to assumed unrealistic moisture conditions and demonstrated that frost cracking patterns deviate from measured fracture systems (Draebing and Mayer, 2021). Erosion rate measurements revealed higher erosion rates at north-facing rockwalls than south-facing rockwalls in the Alps (Coutard and Francou, 1989; Sass, 2005b). Differences between efficacy of freeze-thaw cycles and erosion rates can relate to process

involved in erosion that are not included in frost cracking models such as permafrost or paraglacial processes that need to be taken into account.

Climate change can accelerate or impede frost weathering efficacy in the Alps. Global warming influences temperature elevational patterns in the mountains leading to warming, reduced snow season and stronger temperature fluctuations (Pepin et al., 2022; Pepin et al., 2015; Bender et al., 2020). Our results show that the efficacy of frost weathering is dependent on

number of freeze-thaw cycles and sustained freezing periods regarding sufficient temperatures for frost cracking. Warming at elevations above 3000 m will lead to warmer mean annual rock surface temperatures and a reduced snow duration that will amplify the occurrence of frequent freeze thaw cycles and decrease sustained freezing and result in temperature conditions as currently observed at south-facing rockwalls. At low-elevated rockwalls such as the Dammkar (Fig.1a), warming will decrease the number of freeze-thaw cycles and, therefore, efficacy of frost cracking. The shift of frost weathering will provide more





erosion at high-elevation sites and effect debris-flow activity due to changes of sediment availability (Hirschberg et al., 2021;
Rengers et al., 2020) and therefore changes hazard potential at the valley bottom.

**5 Conclusion**

We quantified frost-induced rock damage on Dachstein limestone by combining X-ray computed micro-tomography (µCT),
acoustic emission (AE) monitoring and frost cracking modelling. To differentiate between potential mechanisms of rock

damage, thermal- and ice-induced stresses were simulated and compared with AE activity. Our study showed that µCT can be
used to quantify frost damage in low porosity (0.1%) rock. Overall, rock damage occurred by propagation of pre-existing
cracks rather than by initiation of new cracks. AE activity, thermal and ice stress models suggest that rock damage is
predominantly caused by a combination of thermal stresses and ice segregation during cooling. In the fully saturated samples,
the occurrence of volumetric expansion cannot be excluded as a potential amplifier of damage. At temperatures below -6 °C,

AE data in combination with frost cracking modelling indicate rock damage driven by ice segregation, while during warming
thermal stress modelling suggests further rock damage by thermal stress. In summary, ice stresses were the dominant factor of
rock damage, however, magnitude of AE activity and frost cracking model results for different saturation regimes differ from
rock damage quantified by µCT, which we interpret as a result of complex crack geometry, saturation assumptions of frost
cracking models and unknown triggers of AE activity. Based on µCT results, frequent freeze-thaw cycles (FT-1) showed

higher cracking efficiencies than sustained freeze-thaw cycles (FT-2). This pattern could be a result of limited water availability
in FT-2 due to the low porosity of our samples. For natural rockwalls, our results indicate higher frost cracking activity in low-
elevation or south-facing rockwalls where frequent freeze-thaw cycles are currently more common. However, our results
contradict the few available erosion measurements. Further testing of the limiting factor of water availability should be
undertaken as modelled ice stresses differ from quantified rock damage in FT-2.


**Data availability**

Data will be uploaded to a data depository when the paper is accepted. Until then, data is available on request from the
corresponding author.

**Author Contribution**

TM, MD, LS and DD designed the experiments. TM, MD and LS conducted the experiments and TM processed the data
with support by MD and LS. TM, MD, LS, VC and DD contributed to manuscript writing and editing.

**Conflict of Interest Statement**

The authors declare that they have no conflict of interest.



**Acknowledgements**

This project has received funding from the European Union's Horizon 2020 research and innovation programme under grant agreement No 101005611 for Transnational Access conducted at Ghent University. The laboratory work was funded by Excite grants FROST (EXCITE_TNA_C1_2022_06) and CRACK (EXCITE_TNA_C2_2022_004), and by German Research Foundation (DR1070/3-1, 426793773). The authors thank Timo Sprenger for his help at collecting rock samples in the field. The Ghent University Special Research Fund (BOF-UGent) is acknowledged to support the Centre of Expertise UGCT (BOF.COR.2022.0009).

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
