# Peer review of "Quantifying frost weathering induced damage in alpine rocks"

_The Cryosphere, 2023_

## Author Comment (AC1)

Reviewer comments (in blue), our response (in black), revised text (in orange)

**Reviewer #2 comments:**

1) I have a few concerns regarding the setup and representativeness of laboratory experiments.

First, Figures 1 and 2 are misleading because the authors show selected alpine rock walls and related temperature time series.

We changed Figures 1 and 2 completely and created a new Figure 1 focusing on experimental setup, samples, and applied temperature cycles.

[Figure]

Figure 1: The design of the experiment was influenced by the methodologies of Hallet et al. (1991) and Mayer et al. (2023), focusing on distinguishing ice segregation as a distinct weathering process from other weathering mechanisms. a) Schematic representation of freezing laboratory setup. We created a linear temperature gradient by cooling three rock samples, each with varying levels of saturation, using a cooling plate positioned at the bottom, while exposing the top of the samples to ambient room temperature conditions. b) 91.5 x 25.5 mm large cylindrical Dachstein limestone samples used for freeze-thaw experiments. c) temperature cycles were implemented for FT-1 and FT-2, along with the

corresponding measurements of rock and cooling plate temperatures. In FT-1, between scan 0 and 1, there was inadequate coupling of the temperature sensor, resulting in excessively high temperature readings. d) temperature isoplots of derived temperature distribution within the sample.

Still, the thermal conditions in the laboratory experiments do not correspond to reality (neither the gradients nor the duration, which are, besides the water availability, crucial for ice segregation).

We deviated from highlighting rockwall conditions in our experimental runs and added in the text: 'With this setup we simulated an closed system that provides a linear temperature gradient and a water body inside the rock samples reflecting simplified natural rockwall conditions.'

Further, the timely changing thermal conditions (i.e., periods with non-linear temperature gradient in the sample) driven by the cooling plate at the bottom are not adequately considered (no uniform temperature gradient from bottom to top and lateral temperature gradient from inside to outside are ignored).

We added in Fig1 d the isoplots for both experimental runs (see above) to show temperature dependencies of the samples. We also added in the method section a more critical statement: 'The surface temperature of the sample has a slight offset compared to the internal temperature at the center, with lower temperatures in the core of the sample. However, we assume this offset as minor due to the high thermal conductivity of limestone (2.4 Wm$^{-1}$K$^{-1}$ (Cermák and Rybach, 1982).'

Second, the number of samples is minimal (only 1 sample/experiment for each condition), does not allow such firm conclusions, and certainly not for a direct implication to the real world.

We weakened our conclusions and fully deleted section 4.4 with interpretations of our findings onto rockwalls. We critically stated the results about saturation impact as sample number was low and saturation levels did change over the experiment. Method Section: 'As saturation influences frost weathering, we used rock samples with an initial saturation of approximately 30, 70 and 100 % categorized as low, partially, and highly saturated, respectively. The samples were saturated by immersing the lower part into a distilled water bath. To prevent air inclusions, we raised slowly the water table until samples were completely immersed, and a constant mass was prevailing (we refer to as highly saturated). Subsequently, samples were dried under atmospheric conditions, weighed until low (30 %) or partially saturation (70 %) was reached. To minimize moisture loss through evaporation, the samples were wrapped in clingfilm. As porosity of the samples is 0.1 %, the level of saturation is inaccurate and provide only a rough estimation. Furthermore, the saturation can change during the experiments due to moisture loss or distribution of rock moisture can alter within the rock samples. We chose the length of the rock samples of 91.5 mm to enable moisture migration towards the sample parts close to the cooling plate where freezing occurred. Due to the low number of samples, we cannot quantify saturation effects on frost weathering, however, our set up enables us to incorporate the ariability of saturation levels occurring in natural rockwalls and to test the consistency frost damage patterns. '

Still, our results highlight that all samples showed the same dependency between initial crack density and crack growth. We can also conclude that all samples of FT-1 showed a higher crack growth compared to FT-2.

Discussion section: 'Based on µCT data, freeze-thaw cycles (FT-1) revealed higher rock damage compared to a sustained freezing-cycle (FT-2) in low-porosity crack-dominated alpine rocks. Final crack growth is affected by initial crack density or pore volume distribution and cannot be compared directly; however, normalized crack growth fraction revealed an increase between 29 and 52 % for FT-1 compared to an increase between 2 and 12 % for FT-2 (Fig. 6a, d). The increase varied between samples of different saturation levels. While the low number of samples prohibit a quantitative analysis of saturation effects on rock damage, our results are consistent and reveal higher damage of FT-1 compared to FT-2 independent of saturation levels (Fig. 6a, d).'

Third, the coupling between the AE sensor and the rock is crucial, but there is no proof of how good it is and if it is comparable between the experiment's start and end. For example, how do you explain the change in the yellow slope in Figure 4c?

We did measure coupling with lead break tests before and after the experiment. We added in the text: 'We performed lead break tests as sample cracks (Eppes et al., 2016) before and after each scan to control sensor coupling and evaluating system performance and wavelength form. Poor coupling of an AE sensor could lead to diminished AE amplitudes, meaning signals of low amplitude might not be detected by the system. To avoid erroneous AE signals stemming from the setup, the system underwent testing without any freezing or temperature alterations.'

We stated a potential coupling shift in the results section:

'Throughout the cycles, a pattern of accumulating AE hits among the samples is evident. Initially, the highly saturated sample accumulated the majority of hits in the first 10 cycles (121 hits). However, there was a notable increase in AE hits for the partly saturated sample, eventually leading to a higher total than the highly saturated sample between 10 and 20 cycles (173 hits). Given the consistent trend observed in both the highly and low saturated samples, a likely shift in the coupling between the AE sensor and the sample is suggested. Consequently, it is probable that the total AE hits for the partly saturated sample were fewer than those for the highly saturated one. Due to two recording interruptions of the AE logger, AE hits for the FT-1 cycle are likely underestimated.'

We critically discussed the change in slope at the yellow line in Figure 4c in the discussion section:

'A potential alteration in the AE sensor's attachment to the rock might affect also signal detection. Although we reattached sensors and conducted lead break tests during the FT-1 cycle, the connection could have changed over time. Given the consistent AE accumulation trend observed in both the highly and low saturated samples, a likely coupling shift at the partially saturated sample is suggested (Fig. 3e), which was not reflected in µCT crack growth data (Fig. 5b, h).'

Further, a direct comparison between two AE time series must be interpreted carefully, and the threshold level might need to be adjusted. For example, if you normalized the summed AE hits in Figure 7, I would expect that they all have a similar pattern (repeating in FT-1, comparable even to FT-2). Therefore, at the moment, the AE results are over-interpreted, and a more in-depth evaluation is required.

We utilized the AE Win software to analyse and filter the incoming AE signals, verifying that no adjustment of the threshold was necessary. We stated this in the method section:

 'The detected AE signals were recorded with a Physical Acoustics micro SHM node. Recorded data were subsequently processed and filtered using Physical Acoustics AEwin software. Due to low background

noises of our setup, we set an initial signal threshold of 30 dB$_{AE}$, which is similar to Mayer et al. (2023) who established a threshold of 35 dB$_{AE}$, due to the presence of stronger background noises. We performed lead break tests as sample cracks (Eppes et al., 2016) before and after each scan to control sensor coupling and evaluating system performance and wavelength form.'

We critically discussed AE logger results in the discussion section:

**'4.1 Critical discussion on AE Monitoring, stress modelling and µCT technique**

Thermal and ice stresses or a combination of these stresses can cause rock damage. We monitored AE as a proxy for cracking as previous stress experiments (Eppes et al., 2016; Hallet et al., 1991) and analysed the timing of AE events in combination with simplified thermal stress and ice stress models to decipher the potential stress source. Our findings indicate a proportional relationship between the number of AE events and rock damage quantified via µCT (Fig. 6a,d) which was also shown by findings of Wang et al. (2020a). However, in our study the highest count of AE hits does not always align with the most visible rock damage (Fig. 6a). Specifically, the partially saturated sample exhibited over 415 AE hits with a normalized crack growth fraction of 47%, while the low saturated sample displayed 180 AE hits alongside a 53% crack growth. In contrast to our setup, Wang et al. (2020a) utilized a rock sample with artificially created macro fractures that predominantly drove the generation of AE signals. The discrepancy in our results might be due to variations in volume growth per crack propagation, potentially causing fewer AE releases with greater porosity growth. Additionally, the distinct responses of our natural rock samples to stress, influenced by slightly varying rock parameters, crack distribution and saturation, could also impact the number of AE hits. A potential alteration in the AE sensor's attachment to the rock might affect also signal detection. Although we reattached sensors and conducted lead break tests during the FT-1 cycle, the connection could have changed over time. Given the consistent AE accumulation trend observed in both the highly and low saturated samples, a likely coupling shift at the partially saturated sample is suggested (Fig. 3e), which was not reflected in µCT crack growth data (Fig. 5b, h). Despite these variables, the partially saturated sample showed before the shift already a higher AE accumulation than the less saturated one, underscoring that AE hits did not completely correlate with crack growth.'

We changed figure 7 and normalized the summed AE hits. We simplified our interpretation and focused with AE hits timing and occurrence. We added in the text: 'The setup of FT-1 enables the development of thermal stresses during cooling and warming of the samples, volumetric expansion alongside the expanding freezing front and ice segregation during freezing conditions (Fig. 7a). In addition, FT-2 favours the development of ice segregation when temperatures are sustained and rock moisture is able to migrate towards the freezing front (Fig. 7a). '

[Figure]

Fig. 7: Cooling phases accumulate more AE hits compared to warming phases, suggesting frost cracking as the main contributor to rock damage. a) Timing examples of potential stress occurrences include thermal stresses during sample cooling and warming, volumetric expansion along with the expanding freezing front, and ice segregation under freezing conditions. Cumulative AE hits are plotted against the bottom rock temperature sensor for b-d) FT-1 and e-g) FT-2.

Fourth, I'm very fascinated by the micro-CT results. I wonder if you saw similar patterns at other locations (e.g., vertically above). Nevertheless, the comparison between the scans seems to be not very sophisticated/quantitative (there are many approaches from photogrammetry for quantifying changes …).

We followed state of the art procedure in μCT analyses which was also done by Cnudde and Boone (2013). We modified the methodology sections to enhance comprehension and introduced a new figure, further contributing to a clearer understanding.

'All subsequent image handling, such as registration, segmentation, and analyses, were performed with Avizo3D Pro (Version 2021.1, ThermoFisher Scientific). In Avizo, a sandbox filter was conducted to bin contrast variations inside the images and match contrast between the single scans. We tuned the

parameters until visually the best result was observed. Therefore, sample voids (pore space) and matrix (sample material) of each image could be separated by thresholding over contrast. We followed the work after Deprez et al. (2020a) and defined a distinguishable feature in the scan image as a minimum spatial resolution of 3 times the voxel size (60 μm). Volume fractions and the expansion of pore space in the sample were assessed using photogrammetry in Avizo. For each image, the software detected and quantified distinctions between void voxel $V_V$ and the matrix voxel $V_M$ which we call crack fraction $cf$ (Fig. 2b). Crack fraction is derived by:

$$cf = \frac{V_V}{V_V + V_M} \tag{1}$$

This approach enabled subsequent comparisons between scans to assess the development of pore space growth in the sample (Fig. 2c,d). The parameter crack fraction was defined by the total amount of segmented pore space (voids) per image in the image stack (cross section) divided by the total amount of segmented sample material (matrix + voids). Due to effects of beam hardening, which result in image distortion at sample heights between 0 and 5 mm and between 19 and 20 mm, our analyses were concentrated on the portion of the rock sample ranging from 5 to 19 mm in height. We quantified crack growth (pore space growth) by comparing the crack fraction per layer after each scan (Fig. 2d and 6a-f).

If growth occurs in every crack or void, this implies that the distribution of initial cracks/voids could either accelerate or decelerate the growth of cracks. Consequently, crack growth cannot be directly compared across samples due to variations in crack distribution. To address this, we adjusted each scan $cf_i$ by its initial crack fraction $cf_0$, allowing for an assessment of crack growth that is independent of the initial crack distribution. The normalized crack fraction, $cf_{norm}$ for each scan is calculated as follows:

$$cf_{norm,i} = \frac{cf_i}{cf_0}, \tag{2}$$

where $i$ represents the scan number. For the purpose of assessing the progression of quantified frost damage both within a single sample and among different samples, we computed the mean of $cf_{norm}$ for each scan.

,

[Figure]

Figure 2: Schematic drawing from scan to crack growth. a) scanned volume from 0 – 20 mm sample height. Due to beam hardening effects (black dashed area) at the edges only the area between 5 - 19 mm sample height was analysed. b) Example scan of one layer (20 µm thick) with example void/crack voxel and matrix voxel derived by Avizio3D Pro. c) Crack fraction derived for each layer over the whole scanned height from 5 - 19 mm by photogrammetry. d) Example of resulting crack growth per cycle with initial crack fraction (blue line).

We also edited Figure 5 for a better understanding of our results.

[Figure]

Figure 4: μCT visualizes crack growth at both cycles FT-1 and -2. a) 3D CT scans before (scan 0) and after the last freeze-thaw cycle (scan 4) of low-saturated rock sample experiencing FT-1 and c) partially saturated rock sample. The initial distribution of cracks/voids is depicted in black, whereas red illustrates solely the isolated growth of these cracks. Example CT scan slices at 8 mm height from the bottom for b) the low saturated sample exposed to FT-1 and d) for the partially saturated samples experiencing FT-2.

Regarding your comment: 'Fourth, I'm very fascinated by the micro-CT results. I wonder if you saw similar patterns at other locations (e.g., vertically above).' In the scanned section of the sample (5 to 19 mm sample height) we did find similar patterns which can be seen in the following figure.

[Figure]

Figure 5: µCT effectively showcases the progression of cracks in our samples, illustrating a steady growth in crack volume that is uniform throughout different heights in the sample and directly correlates with the initial crack volume. a-f) Measured crack fraction (volume of cracks/voids divided by total volume, refer to Fig. 2b) using µCT plotted against rock depth. g-l) Quantified normalized crack fraction by initial crack volume (blue line a-f) plotted against rock depth.

Fives, how do you assess the scalability and transferability to real-world conditions? There is no critical discussion on this with consideration.

We removed Discussion 4.4 and introduced a new discussion that critically examines the implications of our findings.

**'4.4 Implications for alpine rockwalls**

Our results revealed that the presence and arrangement of voids and fractures within rock significantly impact frost damage. We have shown that micro-crack expansion tends to follow pre-existing fractures, extending their width and length (Fig. 4 and 5g-l), leading to a gradual increase in crack size. As a result, samples with a higher crack density experience more severe frost damage. In natural rockwalls, both micro and macro cracks are present, the latter often arising from tectonic forces and/or weathering effects. These fractures play a crucial role for erosion processes as they influence rock cohesion and modify the dynamics, patterns, and locations of geomorphic activities on various spatial and temporal scales (Scott and Wohl, 2019). Studies by Hales and Roering (2009) and Draebing and Mayer (2021) have established a link between frost cracking intensity and the density of fractures, with rockwalls exhibiting more fractures also showing greater evidence of frost cracking. Furthermore, Eppes et al. (2018) have demonstrated through both field and laboratory observations that an increase in the length and quantity of cracks leads to higher long-term erosion rates. Neely et al. (2019) revealed that higher fracture density decreases steepness of cliffs and increases catchment erosion rates. In New Zealand, Clarke and Burbank (2010) showed that bedrock fracturing by geomorphic processes including weathering controls the depths of erosive processes as bedrock landsliding. We infer that upscaling our

findings from micro to macro cracks highlights the connection between erosion and fracture density. However, such extrapolation must consider the scale dependencies and complex fracture interactions influenced by broader geological and environmental factors, including tectonic forces, weathering effects, and variations in material properties, which could significantly modify erosion dynamics beyond micro-scale observations.

Our findings indicate that frost cracking is more effective during freeze-thaw cycles than during prolonged periods of freezing. Matsuoka et al. (1998) indicated that south-facing rockwalls typically undergo more freeze-thaw cycles due to lack of snow cover, whereas those facing north are subject to longer durations of freezing. This leads to the initial assumption that south facing rockwalls would sustain more frost damage, contributing to increased erosion. However, few existing empirical data indicates that erosion rates are actually 2.5 to 3 times (Sass, 2005a) or up to one magnitude (Coutard and Francou, 1989) higher on north-facing rockwalls. Matsuoka et al. (1998) suggested that while freeze-thaw cycles can cause shallow frost damage (up to 0.3 m deep), prolonged freezing can result in more significant frost damage (up to 5 m deep), leading to larger rockfalls. This indicates that the temporal scale of freeze-thaw cycles plays a crucial role in determining weathering and erosion rates, a concept further supported by Matsuoka (2008), who found that short-term freeze-thaw cycles cause minor crack expansion, whereas long-term freezing leads to more substantial crack widening.

Our research suggests a direct correlation between the frequency of AE events and the extent of rock damage, as measured by micro-CT scanning. However, the highest occurrences of AE hits do not consistently correspond to the most significant observable rock damage. This discrepancy could be attributed to ice formation influenced by thermal gradients, as well as changes and aging in the ice, as discussed by Gerber et al. (2023). This insight has implications for studies that employ AE as an indicator for thermal stress-induced cracking (Eppes et al., 2016; Collins et al., 2018) and frost cracking (Amitrano et al., 2012; Girard et al., 2013) in natural rockwalls.

The estimated ice stresses in our simulations may significantly diverge from the actual ice stresses experienced, leading to differences between simulated ice stresses and observed rock damage, particularly in the FT-2 scenario. This mismatch between model predictions and actual frost damage observations could stem from the model's oversimplified representations of crack geometries and rock properties, or the relatively brief duration of sustained freezing in FT-2 when compared to conditions on a real rockwall. Research efforts such as those by Draebing and Mayer (2021) or Sanders et al. (2012) which utilize frost cracking models to assess frost damage, might have overemphasized the effects of frost weathering. Nonetheless, Draebing et al. (2022) showed that frost weathering simulations do correspond with the erosion rates observed on north-facing rockwalls, where extended periods of freezing are more common. '

2) The frost-cracking model strongly depends on parametrization and assumptions. Please make a sensitivity analysis, allowing you to visualize the output probabilistically. Publishing the code would certainly give more confidence in the model – I strongly recommend it!

In the Supplementary Information, we included a sensitivity analysis and provided a discussion on it. We altered the values by 10% to observe the impact on the model predictions for FT-1:

[Figure]

For FT-2:

We added a critical discussion in the discussion section:

[revised manuscript text omitted]

We enlarged the font size across all figures and enhanced the dpi value. The color scheme was selected based on Wong (2011) recommendations, who devised a color scheme that is easily readable by individuals with visual impairments.

Wong, B.: Points of view: Color blindness, Nature Methods, 8, 441-441, 10.1038/nmeth.1618, 2011.

Since most of the figures have already been presented earlier, we will not repeat them here. However, we will introduce the two figures that have not yet been displayed:

[Figure]

[Figure]

4) The manuscript is rather lengthy, especially the introduction. Please reorganize, restructure, and shorten the introduction to avoid repetitions. Section '3 Results' is rather hard to read – please add more explanation and interpretation.

We revised and rearranged the introduction to improve clarity and readability, reducing its length by one third.

**'1 Introduction**

[revised manuscript text omitted]

---

## Author Comment (AC2)

Reviewer comments (in blue), our response (in black), revised text (in orange)

**1. Reviewer comments:**

Problems of laboratory settings: The temperature at the sample bottom was used in the analysis, but this ignores the thermal gradient in the sample. If crack growth occurs near the bottom, the analysis is acceptable. When the sample bottom dropped below the freezing point, perhaps most of the scanned area has not yet been frozen, as the side sensor located just above the scanned area never showed subzero temperatures. Thus, the bottom temperature may be much lower than the temperature at which cracking is most active.

We specified that only the bottom temperature logger was utilized in conjunction with the results from the AE logger in the previous Figure 7. This approach helps differentiate between cooling, steady conditions, and warming phases, effectively representing the thermal gradient. For all other analyses, we considered the entire linear temperature distribution within the rock body. To enhance understanding of the temperature distribution, Figures 1 and 2 have been thoroughly revised to better highlight the experimental framework, sample materials, and the implemented temperature cycles. This is new Figure 1:

[Figure]

[Figure]

Figure 1: The design of the experiment was influenced by the methodologies of Hallet et al. (1991) and Mayer et al. (2023), focusing on distinguishing ice segregation as a distinct weathering process from other weathering mechanisms. a) Schematic representation of freezing laboratory setup. We created a linear temperature gradient by cooling three rock samples, each with varying levels of saturation, using a cooling plate positioned at the bottom, while exposing the top of the samples to ambient room temperature conditions. b) 91.5 x 25.5 mm large cylindrical Dachstein limestone samples used for freeze-thaw experiments. c) temperature cycles were implemented for FT-1 and FT-2, along with the corresponding measurements of rock and cooling plate temperatures. In FT-1, between scan 0 and 1, there was inadequate coupling of the temperature sensor, resulting in excessively high temperature readings. d) temperature isoplots of derived temperature distribution within the sample.

We also added in the method section a critical statement for the temperature distribution:

'The surface temperature of the sample has a slight offset compared to the internal temperature at the center, with lower temperatures in the core of the sample. However, we assume this offset as minor due to the high thermal conductivity of limestone (2.4 $Wm^{-1}K^{-1}$) (Cermák and Rybach, 1982).'

Another problem is the lack of an external water reservoir, which, if present, might supply water continuously and cause progressive crack growth in a sustained freezing condition.

Incorporating an additional water bath could potentially intensify ice segregation, thereby amplifying frost damage and crack propagation. However, our experimental configuration did not permit the construction of a water bath. The selection of the sample size was constrained by the resolution capabilities of the µCT. A larger sample would have made it impossible to attain the desired 20 µm resolution. This limitation in sample size also resulted in restricted space available for cooling equipment and an acoustic emission (AE) logger. The sample's bottom was subjected to cooling, while an AE sensor was mounted on the top. Based on prior experiments using AE loggers in a water bath, we were aware that this setup would be ineffective, as the water bath tends to act as a large resonating chamber, amplifying any background noise. Furthermore, we decided against implementing a top-mounted water bath, considering that the water might simply percolate through the sample due to the characteristic crack density of the limestone samples. Nevertheless, we incorporated a statement in the method section to address this point.

'The setup aimed to enable the migration of water from the unfrozen segment to the frozen one enhancing the potential for frost cracking; however, the sample's low porosity combined with the small size of the unfrozen segment limits the amount of water that can migrate, potentially resulting in an underestimation of frost damage compared to what might be observed in natural conditions.'

And in the discussion section:

'…In contrast, our rock samples exhibit much lower porosities of around 0.1%, offering a substantially smaller water reservoir. This suggests that the water supply in FT-2 may have been also a limiting factor for ice segregation, despite the temperature conditions being conducive to this process (e.g. Walder and Hallet, 1985).….'

Limited number of experimental runs: Only one sample was used for each moisture and freeze-thaw regime (6 samples in total), despite significant variability of crack conditions. In pore-supported rocks,

samples are assumed to be relatively homogeneous, but in crack-supported rocks, samples should be inhomogeneous as indicated in the CT images (Fig. 5) and crack volume profiles (Fig. 6). For such inhomogeneous rocks, multiple samples are required to guarantee the reproducibility and to evaluate the effect of moisture levels. In fact, Figure 6 seems to suggest that crack growth is almost independent of saturation but dependent mainly on the combination of the initial crack volume and saturation.

Indeed, this is the reason why ice segregation is commonly measured in homogenous rock rather than in orogenically stressed, fractured rock. Our aim was to assess the impact of ice segregation on low-porosity alpine rock. Given the brief duration of the experiment, constrained by the available slots for beam time, and the resolution of the µCT, we were uncertain whether our method would yield any observable results in terms of porosity growth. The number of samples we could analyze was constrained by the limited beam time available through our funding. In our discussion section, we critically address the small sample size and draw attention to the varying results. Our focus, however, is on comparing the outcomes of the two experiment runs, FT-1 and FT-2, rather than individual samples. A key discovery from our study is the dependency of crack growth on the initial crack volume.

Therefore, we weakened our conclusions and fully deleted section 4.4 with interpretations of our findings onto rockwalls. We critically stated the results about saturation impact as sample number was low and saturation levels did change over the experiment. 'Based on µCT data, freeze-thaw cycles (FT-1) revealed higher rock damage compared to a sustained freezing-cycle (FT-2) in low-porosity crack-dominated alpine rocks. Final crack growth is affected by initial crack density or pore volume distribution and cannot be compared directly; however, normalized crack growth fraction revealed an increase between 29 and 52 % for FT-1 compared to an increase between 2 and 12 % for FT-2 (Fig. 6a, d). The increase varied between samples of different saturation levels. While the low number of samples prohibit a quantitative analysis of saturation effects on rock damage, our results are consistent and reveal higher damage of FT-1 compared to FT-2 independent of saturation levels (Fig. 6a, d).'

I am also concerned about the accuracy of saturation level, because for the low porosity rocks difference in the weight of water is very small between the three saturation levels. Multiple experimental runs are required also to overcome the inaccuracy of the saturation levels.

We critically discussed the impact of saturation fluctuations in the method sections as well as in the discussion. We lowered interpretations on saturation impact.

Method section:

'As saturation influences frost weathering, we used rock samples with an initial saturation of approximately 30, 70 and 100 % categorized as low, partially, and highly saturated, respectively. The samples were saturated by immersing the lower part into a distilled water bath. To prevent air inclusions, we raised slowly the water table until samples were completely immersed, and a constant mass was prevailing (we refer to as highly saturated). Subsequently, samples were dried under atmospheric conditions, weighed until low (30 %) or partially saturation (70 %) was reached. To minimize moisture loss through evaporation, the samples were wrapped in clingfilm. As porosity of the samples is 0.1 %, the level of saturation is inaccurate and provide only a rough estimation. Furthermore, the saturation can change during the experiments due to moisture loss or distribution of rock moisture can alter within the rock samples. We chose the length of the rock samples of 91.5 mm to enable moisture migration towards the sample parts close to the cooling plate where freezing occurred. **Due to the low number of samples, we cannot quantify saturation effects on frost weathering, however, our set up enables us to incorporate the variability of saturation levels occurring in natural rockwalls and to test the consistency frost damage patterns.**'

Discussion section:

'Based on µCT data, freeze-thaw cycles (FT-1) revealed higher rock damage compared to a sustained freezing-cycle (FT-2) in low-porosity crack-dominated alpine rocks. Final crack growth is affected by initial crack density or pore volume distribution and cannot be compared directly; however, normalized crack growth fraction revealed an increase between 29 and 52 % for FT-1 compared to an increase between 2 and 12 % for FT-2 (Fig. 6a, d). The increase varied between samples of different saturation levels. While the low number of samples prohibit a quantitative analysis of saturation effects on rock damage, our results are consistent and reveal higher damage of FT-1 compared to FT-2 independent of saturation levels (Fig. 6a, d).'

Visualization of crack growth: Crack growth is not visible in CT images (Fig. 5), although significant volume increase (30-50 %) is computed from the scanning results. Do you have more clear CT images showing crack extension/widening, or can you explain the reason for the invisibility?

We adapted Fig. 5 for a better understanding of crack growth. The pictures do visualize crack growth.

[Figure]

Figure 4: µCT visualizes crack growth at both cycles FT-1 and -2. a) 3D CT scans before (scan 0) and after the last freeze-thaw cycle (scan 4) of low-saturated rock sample experiencing FT-1 and c) partially saturated rock sample. The initial distribution of cracks/voids is depicted in black, whereas red

illustrates solely the isolated growth of these cracks. Example CT scan slices at 8 mm height from the bottom for b) the low saturated sample exposed to FT-1 and d) for the partially saturated samples experiencing FT-2.

The gap between the experiments and natural conditions: Instant cooling to -10 degrees used in this study is unusual in natural rockwall conditions. AE events during such a rapid cooling (and warming) are certainly attributable to thermal stress, but is it applicable to weathering in natural conditions?

Thanks for this comment, the reviewer is right and we incorporated a critical evaluation of our freezing rate and its possible influence on frost cracking in our experiment.

'The freezing rate we utilized, 12.5°C per hour at the cooling plate, might surpass those observed in natural rockwall settings, yet it is comparable to the rates employed in earlier freezing studies (Jia et al., 2015; Matsuoka, 1990). The applied freezing rate could amplify frost cracking and result in an overestimation of frost damage.'

We diverged from emphasizing the reflection of rockwall conditions in our experimental runs, a point now mentioned in the text.

'With this setup we simulated a closed system that provides a linear temperature gradient and a water body inside the rock samples reflecting simplified natural rockwall conditions.'

We critically discussed the differences between our experimental setup and natural rockwall conditions in a new discussion point '4.4 Implications for alpine rockwalls'.

'Our results revealed that the presence and arrangement of voids and fractures within rock significantly impact frost damage. We have shown that micro-crack expansion tends to follow pre-existing fractures, extending their width and length (Fig. 4 and 5g-l), leading to a gradual increase in crack size. As a result, samples with a higher crack density experience more severe frost damage. In natural rockwalls, both micro and macro cracks are present, the latter often arising from tectonic forces and/or weathering effects. These fractures play a crucial role for erosion processes as they influence rock cohesion and modify the dynamics, patterns, and locations of geomorphic activities on various spatial and temporal scales (Scott and Wohl, 2019). Studies by Hales and Roering (2009) and Draebing and Mayer (2021) have established a link between frost cracking intensity and the density of fractures, with rockwalls exhibiting more fractures also showing greater evidence of frost cracking. Furthermore, Eppes et al. (2018) have demonstrated through both field and laboratory observations that an increase in the length and quantity of cracks leads to higher long-term erosion rates. Neely et al. (2019) revealed that higher fracture density decreases steepness of cliffs and increases catchment erosion rates. In New Zealand, Clarke and Burbank (2010) showed that bedrock fracturing by geomorphic processes including weathering controls the depths of erosive processes as bedrock landsliding. We infer that upscaling our findings from micro to macro cracks highlights the connection between erosion and fracture density. However, such extrapolation must consider the scale dependencies and complex fracture interactions influenced by broader geological and environmental

factors, including tectonic forces, weathering effects, and variations in material properties, which could significantly modify erosion dynamics beyond micro-scale observations.

Our findings indicate that frost cracking is more effective during freeze-thaw cycles than during prolonged periods of freezing. Matsuoka et al. (1998) indicated that south-facing rockwalls typically undergo more freeze-thaw cycles due to lack of snow cover, whereas those facing north are subject to longer durations of freezing. This leads to the initial assumption that south facing rockwalls would sustain more frost damage, contributing to increased erosion. However, few existing empirical data indicates that erosion rates are actually 2.5 to 3 times (Sass, 2005) or up to one magnitude (Coutard and Francou, 1989) higher on north-facing rockwalls. Matsuoka et al. (1998) suggested that while freeze-thaw cycles can cause shallow frost damage (up to 0.3 m deep), prolonged freezing can result in more significant frost damage (up to 5 m deep), leading to larger rockfalls. This indicates that the temporal scale of freeze-thaw cycles plays a crucial role in determining weathering and erosion rates, a concept further supported by Matsuoka (2008), who found that short-term freeze-thaw cycles cause minor crack expansion, whereas long-term freezing leads to more substantial crack widening.

Our research suggests a direct correlation between the frequency of AE events and the extent of rock damage, as measured by micro-CT scanning. However, the highest occurrences of AE hits do not consistently correspond to the most significant observable rock damage. This discrepancy could be attributed to ice formation influenced by thermal gradients, as well as changes and aging in the ice, as discussed by Gerber et al. (2023). This insight has implications for studies that employ AE as an indicator for thermal stress-induced cracking (Eppes et al., 2016; Collins et al., 2018) and frost cracking (Amitrano et al., 2012; Girard et al., 2013) in natural rockwalls.

The estimated ice stresses in our simulations may significantly diverge from the actual ice stresses experienced, leading to differences between simulated ice stresses and observed rock damage, particularly in the FT-2 scenario. This mismatch between model predictions and actual frost damage observations could stem from the model's oversimplified representations of crack geometries and rock properties, or the relatively brief duration of sustained freezing in FT-2 when compared to conditions on a real rockwall. Research efforts such as those by Draebing and Mayer (2021) or Sanders et al. (2012) which utilize frost cracking models to assess frost damage, might have overemphasized the effects of frost weathering. Nonetheless, Draebing et al. (2022) showed that frost weathering simulations do correspond with the erosion rates observed on north-facing rockwalls, where extended periods of freezing are more common. '

Furthermore, the experimental condition lacks external water supply (i.e., moisture source is confined within the rock sample), while in natural conditions water can be supplied from more distant areas.

Certainly, the absence of an external water source could reduce ice segregation and consequently the growth of porosity. Nonetheless, given the relatively brief duration of our experiment and the improbable presence of a water bath in a natural rockwall, we believe that water can migrate towards the freezing front in our samples. In a larger rockwall, water migration towards the surface might be quicker along larger cracks, potentially resulting in more effective ice segregation over an extended period. As a result, compared to actual rockwall conditions, our findings might be conservative or understated.

Incorporating an additional water bath could potentially intensify ice segregation, thereby amplifying frost damage and crack propagation. However, our experimental configuration did not permit the construction of a water bath. The selection of the sample size was constrained by the resolution capabilities of the µCT. A larger sample would have made it impossible to attain the desired 20 µm resolution. This limitation in sample size also resulted in restricted space available for cooling equipment and an acoustic emission (AE) logger. The sample's bottom was subjected to cooling, while an AE sensor was mounted on the top. Based on prior experiments using AE loggers in a water bath, we were aware that this setup would be ineffective, as the water bath tends to act as a large resonating chamber, amplifying any background noise. Furthermore, we decided against implementing a top-mounted water bath, considering that the water might simply percolate through the sample due to the characteristic crack density of the limestone samples. Nevertheless, we incorporated a statement in the method section to address this point.

'The setup aimed to enable the migration of water from the unfrozen segment to the frozen one enhancing the potential for frost cracking; however, the sample's low porosity combined with the small size of the unfrozen segment limits the amount of water that can migrate, potentially resulting in an underestimation of frost damage compared to what might be observed in natural conditions.'

And in the discussion section:

'…In contrast, our rock samples exhibit much lower porosities of around 0.1%, offering a substantially smaller water reservoir. This suggests that the water supply in FT-2 may have been also a limiting factor for ice segregation, despite the temperature conditions being conducive to this process (e.g. Walder and Hallet, 1985).….'

Section 1

Figure 2a: Are the minor grids on the time axis shown at an interval of 12 days?

We revised Figures 1 and 2 and created a new figure.

Figure 2, caption: Are these curves drawn on the basis of subdaily-scale temperatures or daily mean values? Please add the recording intervals.

We revised Figures 1 and 2 and created a new figure.

L101-102: …laboratory freeze-thaw tests never demonstrated… →Duca et al. (2014) showed crack formation in granite, although they simulated not freeze-thaw but sustained freezing. Wang et al. (2020) also showed F-T induced crack extension in fractured granite with AEs and CT scan.

We rephrased the sentence and implemented the studies of Duca et al. (2014) and Wang et al. (2020) at several locations in our study: '…. Consequently, laboratory studies have adopted other indicators such as AE signals (Hallet et al., 1991; Mayer et al., 2023; Maji and Murton, 2021; Duca et al., 2014), frost heave or crack expansion (Murton et al., 2006; Draebing and Krautblatter, 2019), alterations in mechanical properties like p-wave velocity, Youngs' Modulus, uniaxial strength or porosity (Whalley et al., 2004; Matsuoka, 1990; Jia et al., 2015; Eslami et al., 2018; Prick, 1997), and frost cracking simulations (Mayer et al., 2023; Murton et al., 2006) to estimate its impact.'

'…In contrast, X-ray computed micro-tomography (µCT) enables the quantification of material damage (Cnudde and Boone, 2013; Withers et al., 2021) and was previously applied to track frost cracking damage in high-porosity rocks (De Kock et al., 2015; Deprez et al., 2020a; Maji and Murton, 2020; Dewanckele et al., 2013) or assess post -experimental frost damage along artificial cracks in low-porosity rocks (Wang et al., 2020a; Wang et al., 2020b) exposed to frequent freeze-thaw cycles.'

'….. This is further corroborated by microscopic analyses by Gerber et al. (2022), who confirmed that such pressure could indeed facilitate pore or crack expansion. Wang et al. (2020b), utilizing µCT with a 35 µm resolution, found that pore space expansion occurs exclusively at or within crack tips, without evidence of new crack formation, only propagation.'

L105: while monitoring… →with monitoring…?

Text was rephrased.

Section 2

L113: 10 X 5 cm →10 cm in length and 5 cm in diameter?

We edited the sentenced to be more clear: '… From boulder one, we drilled three cylindrical samples with 10 cm (in length) x 5 cm (in diameter)….'

L135-136: What is the accuracy of the weight? In 0.1 % porosity and 9.2x2.6 cm sample, my simple estimation indicates that weight of water is 0.13 g, 0.09 g and 0.04 g, respectively for 100, 70 and 30 % sample. How did you control such a small difference in water content?

The accuracy of the scale was 0.005 g. We added a critical statement on how the low porosity of our samples can shift saturation levels especially over experiment time: '…To minimize moisture loss through evaporation, the samples were wrapped in clingfilm. As porosity of the samples is 0.1 %, the level of saturation is inaccurate and provide only a rough estimation. Furthermore, the saturation can change during the experiments due to moisture loss or distribution of rock moisture can alter within the rock samples. We chose the length of the rock samples of 91.5 to enable moisture migration towards the sample parts close to the cooling plate where freezing occurred. Due to the low number of samples, we cannot quantify saturation effects on frost weathering, however, our set up enables us to incorporate of variability of saturation levels occurring in natural rockwalls and to test the consistency frost damage patterns.'

L138: What kind of material does compose the insulating holder?

We used Extruded Polystyrene Foam.

L140-141: The definition of 'open system': Water supply is confined within the rock sample (lacking external water supply), while in natural conditions water can be supplied from more distant areas? In this respect, this experiment might simulate a 'closed system' condition.

We changed the misleading phrase 'open system' to 'closed system': 'With this setup we simulated a closed system that provides a linear temperature gradient and a water body inside the rock samples reflecting simplified natural rockwall conditions.'

L142-143: Regarding the temperature cycle used in FT-1: 1) Did rock samples thaw completely during such a very short period? Do you have any evidence? 2) Can the results from such an extreme temperature condition be applied to natural rockwalls?

The updated Figure 1d (see below), presented as an isoplot, illustrates the temperature distribution, indicating that the rocks have thawed completely.

[Figure]

We rephrased and highlighted less the comparison to natural rockwall conditions: '…In this study, we exposed low-porosity, high-strength Dachstein limestone to frequent diurnal and seasonal sustained freeze-thaw cycles during laboratory freezing experiments. We monitor acoustic emission events during the experiments and modelled thermal and ice-induced stresses and applied µCT to pre- and post-stressed rocks to quantify and track crack propagation and to assess frost cracking efficacy of different freeze-thaw cycles.'

L201: in between →between?

Changed to 'between'.

L215, Eq. (2): Please define n and t in the text.

We added a definition for n and t: '…After Walder and Hallet (1985) shear modulus $G$ and Poisson' ratio $v$ determine how the penny shaped crack is deformed elastically into an oblate ellipsoid when ice pressure is applied and be described for very thin cracks (w<<c) to

$$\frac{w(n,t)}{c(n,t)} = \frac{4}{\pi}\left(\frac{1-v}{G}\right)p_i.$$
(4)

where *n* represents the incremental depth (1 cm), and *t* denotes the incremental timing (1 min).'

L216: The crack… propagates crack growth V →The crack… propagates at a growth rate V?

We changed the sentence to: 'The crack finally breaks subcritical inelastically at the tip (mode I type) and propagates at a growth rate V…'

L224: 37.1 m s-1: According to Table 1, gamma is dimensionless.

We changed the typo. Gamma is dimensionless.

Section 3

L242: The upper temperature sensor →'The top temperature sensor' may be better.

We changed the sentence: '…while the top sensor consistently registered 15 to 24 °C.'

L243: 100% sample →100% saturated sample

We changed the saturation labels for the samples to: 'As saturation influences frost weathering, we used rock samples with an initial saturation of approximately 30, 70 and 100 % categorized as low, partially, and highly saturated, respectively.'

L244-248: To identify AE events due to thermal stress, it may be important to distinguish AEs during rapid temperature change and those during slow change.

We attempted to quantify the impact and timing of thermal stress through modeling, providing insights into its contribution to rock damage. The correlation between AE hits and thermal stress can be accurately established during non-freezing phases. However, during freezing periods, frost cracking also contributes to AE hits, making them indistinguishable from those caused by thermal stress alone.

L250-252: If the potential AE events during the data gap period is taken into account, the highest AE activity could have occurred between scan 2 and 3 in the 70 % sample?

We added a critical comment about the possibility of a coupling change between AE sensor and rock leading to a enhanced AE hits accumulation for the partially (70 %) rock sample. 'Thermal and ice stresses or a combination of these stresses can cause rock damage. We monitored AE as a proxy for cracking as previous stress experiments (Eppes et al., 2016; Hallet et al., 1991) and analysed the timing of AE events in combination with simplified thermal stress and ice stress models to decipher the potential stress source. During the FT-1 cycle, AE sensors were reattached following each scan. Despite conducting lead break tests to verify the sensor-sample coupling, there is a possibility that the connection between the AE sensor and the sample altered over time. Given the consistent AE

accumulation trend observed in both the highly and low saturated samples, a likely shift in the coupling between the AE sensor and the partially sample is suggested (Fig. 3e) as we do not see a shift in crack growth by µCT (Fig. 5b, h). On the other hand, if the coupling was insufficient before, it may have resulted in reduced amplitudes of detected hits, leading to a diminished capture of signals due to weaker signals not exceeding the minimum threshold. As a result, our data may not fully account for all AE signals that were generated and should be interpreted with care.'

L255 (Figure 4, caption): c-d) Please explain dots (each event?) and lines (cumulative?).

We rephrased the caption: 'Figure 3: FT-1 resulted in significantly more AE hits than FT-2, as indicated by thermal stress models but not reflected in frost cracking models where FT-2 showed higher predicted ice stresses. More AE hits were recorded during freezing phases compared to non-freezing ones. a-b) Respectively measured rock and cooling plate temperatures, c-d) recorded AE hits (coloured dots) and cumulative AE hits (coloured lines), and e-f) modelled thermal and ice stresses at bottom temperature sensor plotted against time for FT-1 and FT-2 with AE hits from all samples. The dashed black line highlights cooling plate temperatures according to the controller while black lines indicated measured plate temperature (cooling plate sensor was attached after first 5 cycles). The temperature offset between scan 0 and 1 during FT-1 (a) is a result of poor connectivity of the bottom temperature sensor. Blue backgrounds highlight periods when bottom rock samples were exposed to temperatures below 0 °C.'

L258 (Figure 4, caption): Blue rectangles →'Blue backgrounds' or 'colored backgrounds' may be more appropriate?

We changed the caption accordingly: 'Blue backgrounds highlight periods when bottom rock samples were exposed to temperatures below 0 °C. '

L262: Do you calculate 'thermal- and ice-induced stresses' at the bottom (highest stress?) or at the scanned depth (lower stress)? Is the ice stress given by which equation?

We modelled thermal- and ice-induced stresses at the bottom of the samples to show the highest stress. We rephrased the caption to be clearer: 'Figure 3: FT-1 resulted in significantly more AE hits than FT-2, as indicated by thermal stress models but not reflected in frost cracking models where FT-2 showed higher predicted ice stresses. More AE hits were recorded during freezing phases compared to non-freezing ones. a-b) Respectively measured rock and cooling plate temperatures, c-d) recorded AE hits (coloured dots) and cumulative AE hits (coloured lines), and e-f) modelled thermal and ice stresses at bottom temperature sensor plotted against time for FT-1 and FT-2 with AE hits from all samples. The dashed black line highlights cooling plate temperatures according to the controller while black lines indicated measured plate temperature (cooling plate sensor was attached after first 5 cycles). The temperature offset between scan 0 and 1 during FT-1 (a) is a result of poor connectivity of the bottom temperature sensor. Blue backgrounds highlight periods when bottom rock samples were exposed to temperatures below 0 °C.'

Ice stress $p_i$ is derived through equation 4: 'After Walder and Hallet (1985) shear modulus $G$ and Poisson' ratio $v$ determine how the penny shaped crack is deformed elastically into an oblate ellipsoid when ice pressure $p_i$ is applied and be described for very thin cracks (w<<c) to

$$\frac{w(n,t)}{c(n,t)} = \frac{4}{\pi}\left(\frac{1-v}{G}\right)p_i.$$
(4)

where $n$ represents the incremental depth (1 cm), and $t$ denotes the incremental timing (1 min).'

 Use consistent expression as '100 % saturated, 70% saturated…' instead of 'fully saturated, partly saturated…'.

We changed saturation labels to: 'As saturation influences frost weathering, we used rock samples with an initial saturation of approximately 30, 70 and 100 % categorized as low, partially, and highly saturated, respectively.'

 You wrote 'volume changes occur in form of crack growth (Fig. 5)', but crack growth is not visible in Fig. 5. Do you have more clear CT images?

We edited Figure 5 to be clearer where crack growth is visible:

[Figure]

Figure 4: μCT visualizes crack growth at both cycles FT-1 and -2. a) 3D CT scans before (scan 0) and after the last freeze-thaw cycle (scan 4) of low-saturated rock sample experiencing FT-1 and c) partially saturated rock sample. The initial distribution of cracks/voids is depicted in black, whereas red illustrates solely the isolated growth of these cracks. Example CT scan slices at 8 mm height from the

bottom for b) the low saturated sample exposed to FT-1 and d) for the partially saturated samples experiencing FT-2.

We also created a new figure for the µCT method.

[Figure]

Figure 2: Schematic drawing from scan to crack growth. a) scanned volume from 0 – 20 mm sample height. Due to beam hardening effects (black dashed area) at the edges only the area between 5 - 19 mm sample height was analysed. b) Example scan of one layer (20 µm thick) with example void/crack voxel and matrix voxel derived by Avizio3D Pro. c) Crack fraction derived for each layer over the whole scanned height from 5 - 19 mm by photogrammetry. d) Example of resulting crack growth per cycle with initial crack fraction (blue line).

L283-284: The phrase 'crack growth revealed…' seems inappropriate. Do you mean "crack growth positively correlated with initial crack volume"?

We rephrased the whole section. See comment L286.

L286: Insert 'and' between the two adverbs (i.e., uniformly and independently)?

The new rephrased section: 'Conducted with a spatial resolution of 60 µm, the µCT scans indicated that the majority of the pore volume in our samples consisted of cracks, with changes in volume manifesting

as crack expansion (Fig. 4). The expansion of cracks was found to be consistent throughout the height of the sample, indicating that the growth of cracks was uniform from the topmost scanned section (19 mm) to the lowest (6 mm), However, there was a discernible positive relationship between initial crack and subsequent crack growth (Fig. 5a-c). Crack fraction (calculated as crack/void volume relative to total volume, refer to Fig. 2b and Eq. 1) in the initial scan (scan 0) before freezing exhibited variations from $0.023_{-0.009}^{+0.019}$ in the low saturated rock, $0.006_{-0.002}^{+0.005}$ in the partially saturated rock, and $0.007_{-0.002}^{+0.005}$ in the highly saturated rock. The final (scan 4) crack fraction was $0.031_{-0.009}^{+0.028}$ (an increase of 35 %) for the low saturated rock, $0.008_{-0.004}^{+0.006}$ (an increase of 33 %) for the partially saturated rock, $0.011_{-0.003}^{+0.006}$ (an increase of 51 %) for the highly saturated sample.'

L289: Regarding 'the amount of crack volume growth per scan varies with saturation', Figure 6 implies that crack growth is almost independent of saturation, but dependent mainly on the combination of the initial crack volume and saturation.

We rephrase the whole section: 'To mitigate the influence of initial crack distribution on crack growth analysis, we normalized the growth of crack volume for each sample relative to its initial crack fraction (see Eq. 2). This normalization showed that crack growth was consistent throughout the sample's height, though the extent of growth varied with saturation levels (Fig. 5g-i). Scan 4 revealed a final mean normalized crack growth of 1.34 ±0.8 (equivalent to 34% more crack volume than the initial value) for the low saturated sample, 1.29 ±0.11 (29% increase) for the partially saturated sample, and 1.52 ±0.13 (52% increase) for the highly saturated sample. The mean normalized increase in crack volume between scans was 0.8 ±0.03 (8% increase) for the low saturated sample, 0.07 ±0.04 (7% increase) for the partially saturated sample, and 0.13 ±0.05 (13% increase) for the highly saturated sample.'

L290-292: 30-50 % crack growth is significant, but why is such a crack growth unclear in Figure 5?

See answer for comment L282.

Comment on Section 3.2.1: You may also refer to the acceleration of crack growth for the 70 % sample in the later period and deceleration for the 100 % sample, which seem to be consistent with AE activity?

The comparison between measured crack growth by μCT and AE activity is challenging as AE activity did not always follow the highest crack growth. However, in general trend between AE activity and crack growth is visible. We highlighted this in the discussion section: 'Our findings indicate a proportional relationship between the number of AE events and rock damage quantified via μCT (Fig. 6a,d). However, the highest count of AE hits does not always align with the most visible rock damage (Fig. 6a). Specifically, the partially saturated sample exhibited over 415 AE hits with a normalized crack growth fraction of 47%, while the low saturated sample displayed 180 AE hits alongside a 53% crack growth. This discrepancy might be due to variations in volume growth per crack propagation, potentially causing fewer AE releases with greater porosity growth. Additionally, the distinct responses of our natural rock samples to stress, influenced by slightly varying rock parameters, crack distribution and saturation, could also impact the number of AE hits. A change in coupling between the AE sensor and the rock may also impact the number of signals being collected. Nonetheless, even when considering potential changes, such as a coupling change that may have occurred at scan 2 for the partially saturated sample during FT-1 (Fig. 3c), this sample still registered more hits than the less

saturated one. This leads to the consistent conclusion that AE hits did not completely correlate with crack growth.'

Figure 5: If crack extension/widening is visible, please indicate it with an arrow.

See answer for comment on L282.

L314-315: This result suggests that, also in FT-2, the initial crack volume is a more important factor than the saturation?

Yes, we pointed out that the initial crack volume is the driving factor for crack growth. The impact of saturation is difficult to state as the low porosity and a potential saturation loss due to evaporation may have shifted saturation levels. We accounted this in the text: 'Considering the limited number of samples, our experiments provide an initial indication that, based on the activity observed through acoustic emissions and μCT, higher saturation levels appear to enhance the extent of frost-induced damage to the rock. Understanding the effects of saturation on the growth of induced porosity is complex, as factors like evaporation and inherent low porosity led to great reductions in saturation over the experiment run (see also section 2.2). Nevertheless, μCT findings from our experiment indicate an increased crack growth for the highly saturated rock samples during both FT-1 and FT-2 phases, culminating also the highest frequency of AE events. In contrast, the low and partially saturated samples showed less crack growth and AE hit accumulation in FT-1 and FT-2 which differs to findings of Mayer et al. (2023) who suggested no lower moisture boundary for frost cracking under the perquisite of available water within short distance (0.4 m).  However, our setup did not contain an external water bath like Mayer et al. (2023) which may not provide sufficient water for ice segregation in our low-saturated samples. Prick (1997) noted water migration in limestone samples similar in size, with porosities ranging from 26 to 48.2 %. In contrast, our rock samples exhibit much lower porosities of around 0.1%, offering a substantially smaller water reservoir.'

 Section 4

L324-326: The sentence 'This pattern...' is redundant. Perhaps it can be rewritten more concisely.

We rewrote the whole section.

L326-327: Again, Figure 5 does not show crack growth clearly.

See answer on comment L282.

L331: Who are 'the authors': the authors of this paper, all authors cited in the preceding sentences, or only De Kock et al.? You also used 'the authors' in L389, L444 and L455 probably for those of the reference just cited, while 'the authors' in Acknowledgements seem to indicate those of this paper. Please keep consistency.

We changed all sentences including 'the authors' and only kept it in the Acknowledgements.

L351: Correct grammar such that 'Prick's empirical findings suggested that…' or 'Prick empirically found that…'.

We rephrased the sentence to: 'Prick (1997) noted water migration in limestone samples similar in size, with porosities ranging from 26 to 48.2 %.'

L354: 'low-porosity' instead of 'low porous'?

We deleted the whole section.

L359-360: But when the sample bottom dropped below the freezing point, perhaps most of the scanned area has not yet been frozen? Thus, a possibility that volumetric expansion in the scanned occurs later cannot be ruled out?

Volumetric expansion can only be definitively excluded when the temperatures within the samples remain steady, a condition met in FT-2 but only momentarily in FT-1. We have removed this section.

L362-364: This interpretation is acceptable for the present experiment, but does this result based on 'instant cooling (favorable for thermal cracking)' is unlikely applicable to natural rockwall conditions?

We removed this section and added a critical statement about the freezing rate: 'The freezing rate we utilized, 12.5°C per hour at the cooling plate, might surpass those observed in natural rockwall settings, yet it is comparable to the rates employed in earlier freezing studies. (Jia et al., 2015; Matsuoka, 1990). The applied freezing rate could amplify frost cracking and result in an overestimation of frost damage.'

L366-368: But the location of crack growth may not be at the bottom of rock sample where the temperature is recorded. Since the side sensor located just above the scanned area never showed subzero temperatures (Fig. 5), most of the scanned (cracked) area never reached below -6 degrees?

We have restructured this section and no longer draw comparisons between concrete temperature and crack growth. Utilizing the isoplot, as mentioned in the comment on L142, to examine the temperature distribution within the rock body, it was observed that the area scanned remained frozen throughout the cycles.

L369: For the above reason, the result presented here cannot be compared with Hallet et al. (1990)?

We have restructured this section and no longer draw comparisons to Hallet et al. (1990).

L397-398: This sentence seems misleading. If you compare the results per F-T cycle (i.e., summed AE hits or crack volume fraction divided by the number of F-T cycles) between the two conditions, sustained freezing seems to have higher efficacy. Furthermore, F-T cycles with instant cooling to -10 degrees may be unusual in natural conditions. Freeze-thaw tests with milder cooling bay be preferable to apply to natural conditions.

We fully rephrased this section and critically discussed the given constraints in our experiment on FT-1 and FT-2.

**'4.3 Efficacy of freeze-thaw and sustained freezing cycles**

[revised manuscript text omitted]

L405: You may add Duca et al. (2014) and Wang et al. (2020) for the examples of laboratory frost cracking.

See comment on L101.

L409-410: I cannot understand the sentence 'This pattern…'.

We revised this section, focusing on the discrepancies between the accumulation of AE hits and the observed growth in porosity. The highest number of AE hits does not consistently correlate with the most significant frost damage. This has led us to establish a new critical point for discussion:

'4**.1 Critical discussion on AE Monitoring, stress modelling and µCT technique**

'Thermal and ice stresses or a combination of these stresses can cause rock damage. We monitored AE as a proxy for cracking as previous stress experiments (Eppes et al., 2016; Hallet et al., 1991) and analysed the timing of AE events in combination with simplified thermal stress and ice stress models to decipher the potential stress source. Our findings indicate a proportional relationship between the number of AE events and rock damage quantified via µCT (Fig. 6a,d) which was also shown by findings of Wang et al. (2020). However, in our study the highest count of AE hits does not always align with the most visible rock damage (Fig. 6a). Specifically, the partially saturated sample exhibited over 415

AE hits with a normalized crack growth fraction of 47%, while the low saturated sample displayed 180 AE hits alongside a 53% crack growth. In contrast to our setup, Wang et al. (2020) utilized a rock sample with artificially created macro fractures that predominantly drove the generation of AE signals. The discrepancy in our results might be due to variations in volume growth per crack propagation, potentially causing fewer AE releases with greater porosity growth. Additionally, the distinct responses of our natural rock samples to stress, influenced by slightly varying rock parameters, crack distribution and saturation, could also impact the number of AE hits. A potential alteration in the AE sensor's attachment to the rock might affect also signal detection. Although we reattached sensors and conducted lead break tests during the FT-1 cycle, the connection could have changed over time. Given the consistent AE accumulation trend observed in both the highly and low saturated samples, a likely coupling shift at the partially saturated sample is suggested (Fig. 3e), which was not reflected in µCT crack growth data (Fig. 5b, h). Despite these variables, the partially saturated sample showed before the shift already a higher AE accumulation than the less saturated one, underscoring that AE hits did not completely correlate with crack growth.'

L413-414: I think that 'eight times higher ice stresses in the sustained freezing' seems consistent with higher cracking activity for one F-T cycle (see the above comment on L397-398).

See our answer to L397-398.

L421-422: I support this interpretation. If you provide water from the top during freezing, ice segregation may occur more effectively (cf. Duca et al., 2014)?

See comment on L101.

Figure 8, label of the horizontal axis: What is 'Summed Ice Thermal'? Do you mean 'Summed Thermal Stress'?

Indeed. We changed the label to 'Summed Thermal Stress (MPa)'.

L445: The phrase 'an increase of mean annual rock temperatures...' may lead to misunderstanding that warming progresses year by year. 'Higher temperature' may be more appropriate.

This section has been removed.

L465-468: Time scale for warming should also be considered: e.g., can 1-2 degrees of warming lead to significant changes in snow cover and number of freeze-thaw cycles above 3000m?

Indeed, but we removed this section.

L470: Do you mean 'increased debris-flow activity due to increased sediment availability at high elevation sites'?

This section has been removed.